# Cyclic muscle contractions reinforce the actomyosin motors and mediate the full elongation of *C. elegans* embryo

**Anna Dai, Martine Ben Amar\***

Laboratoire de Physique de l'Ecole normale supérieure, ENS, Université PSL, CNRS, Sorbonne Université, Université Paris Cité, Paris, France

**Abstract** The paramount importance of mechanical forces in morphogenesis and embryogenesis is widely recognized, but understanding the mechanism at the cellular and molecular level remains challenging. Because of its simple internal organization, *Caenorhabditis elegans* is a rewarding system of study. As demonstrated experimentally, after an initial period of steady elongation driven by the actomyosin network, muscle contractions operate a quasi-periodic sequence of bending, rotation, and torsion, that leads to the final fourfold size of the embryos before hatching. How actomyosin and muscles contribute to embryonic elongation is investigated here theoretically. A filamentary elastic model that converts stimuli generated by biochemical signals in the tissue into driving forces, explains embryonic deformation under actin bundles and muscle activity, and dictates mechanisms of late elongation based on the effects of energy conversion and dissipation. We quantify this dynamic transformation by stretches applied to a cylindrical structure that mimics the body shape in finite elasticity, obtaining good agreement and understanding of both wild-type and mutant embryos at all stages.

## eLife assessment

Using continuum theory of elastic solids the authors present evidence that periodic muscle contraction leads to elongation of *C. elegans* embryos by storing elastic energy that is subsequently released by extending the embryo's long axis. This **important** finding could apply to other developmental processes and be exploited in soft robotics. The presented evidence is **convincing** on the phenomenological level adopted in the work. How bending energy is converted into elongation on a more microscopic level remains to be worked out.

## Introduction

Mechanical stresses play a critical role during embryogenesis, affecting complex biological tissues such as the skin, the brain, and the interior of organs (*Goriely, 2017*). There are many studies in recent decades, for example, the pattern of the intestine (*Coulombre and Coulombre, 1958*; *Hannezo et al., 2011*; *Li et al., 2011*; *Ben Amar and Jia, 2013*; *Shyer et al., 2013*), the brain cortex (*Toro and Burnod, 2005*; *Goriely et al., 2015*; *Ben Amar and Bordner, 2017*; *Tallinen et al., 2016*), and the circumvolutions of the fingerprints (*Kücken and Newell, 2005*; *Ciarletta and Ben Amar, 2012*), which have been interpreted as the result of compressive stresses generated by the growth that occurs a few months after fertilization in humans. The superposition of mechanical stresses, cellular processes (e.g. division, migration), and tissue organization is often too complex to identify and quantify. For example, independent of growth, stresses can be generated by different molecular motors, among which myosin II, linked to actin filaments, is the most abundant during epithelial cell morphogenesis

\*For correspondence:
benamar@lps.ens.fr

**Competing interest:** The authors declare that no competing interests exist.

**Figure 1.** Schematic diagram of embryonic development of *C. elegans*. (**A**) Overview of *C. elegans* embryonic development. Three epidermal cell types are found around the circumference: dorsal, ventral, and seam cells. (**B**) Schemes showing a *C. elegans* cross-section of the embryo. Small yellow arrows in the left image indicate the contraction force that occurred in the seam cell. Four muscle bands under the epidermis and actin bundles surround the outer epidermis.

(*Vicente-Manzanares et al., 2009*) and cell motility (*Cowan and Hyman, 2007*; *Olson and Sahai, 2009*; *Palumbo et al., 2022*). Spatial distribution and dynamics of myosin II strongly influence the morphogenetic process (*Levayer and Lecuit, 2012*; *Lv et al., 2022*), as demonstrated for *Drosophila* (*Bertet et al., 2004*; *Blankenship et al., 2006*; *Saxena et al., 2014*; *Shindo and Wallingford, 2014*) and also for *C. elegans* embryos (*Priess and Hirsh, 1986*; *Gally et al., 2009*; *Ben Amar et al., 2018*).

Embryonic elongation of *C. elegans* before hatching provides an attractive model of matter reorganization in the absence of growth. It occurs after ventral enclosure and takes about 240 min to transform the lima bean-shaped embryo into the final elongated worm form: the embryo elongates four times along the anterior/posterior axis by *McKeown et al., 1998*. The short lifespan of the egg before hatching and its transparency make this system ideal for studying the forces that exist at or near the cortical epithelium. However, in contrast to *Drosophila* and zebrafish embryonic development, there is no cell migration, cell division, or significant change in embryonic volume (*Sulston et al., 1983*; *Priess and Hirsh, 1986*), only remarkable epidermal elongation drives the entire morphogenetic process of *C. elegans* in the post-enclosure period. There are two experimentally identified driving forces for the elongation: the actomyosin contractility in the seam cells of the epidermis, which appear to persist throughout the whole elongation process, and the muscle activity beneath the epidermis, which begins after the 1.7~1.8-fold stage. The transition is well defined, because the muscle involvement makes the embryo rather motile, and any physical experiments such as laser fracture ablation of the epidermis, which could be performed and achieved in the first period (*Vuong-Brender et al., 2017a*), become difficult. Consequently, the elongation process of *C. elegans* could be divided into two stages: the early elongation and the late elongation, depending on the muscle activation, and *Figure 1A* shows the whole elongation process (*Vuong-Brender et al., 2017a*). Previously, the role of the actomyosin network in the seam cells during the *C. elegans* early elongation was investigated (*Ben Amar et al., 2018*). Based on the geometry of a hollow cylinder composed of dorsal, ventral, and seam cells, a model including the pre-stress responsible for the enclosure, the active compressive ortho-radial stress combined with the passive stress quantitatively predicts the elongation, but only up to ~70% of the initial length.

For the late elongation phase, the actomyosin network and the muscles together drive the full elongation during this phase (*Vuong-Brender et al., 2017a*). The muscles play an important role and mutants with muscle defects are unable to complete the elongation process, even though the

actomyosin network functions normally (*Lardennois et al., 2019*). *Figure 1B* shows a schematic image of the *C. elegans* body (*Zhang et al., 2011*) with four rows of muscles, two of which are under the dorsal epidermis and the other two are under the ventral epidermis. As observed in vivo, *C. elegans* exhibits systematic rotations accompanying each contraction (*Yang, 2017*), and deformations such as bending and twisting. Eventually, the *C. elegans* embryo will undergo an elongation of 1.8-fold to fourfold. As the activated muscle contracts on one side, the contractile forces are transmitted to the epidermis on that side. Since striated muscles can only perform cycles of contraction and relaxation, their action will tend to reduce the length of the embryo. Therefore, it is necessary to understand how the embryo elongates during each contraction and how the muscle contractions are coupled to the actomyosin activity. This work aims to answer this paradox within the framework of finite elasticity without invoking cellular plasticity and stochasticity, which cannot be considered as driving forces. In addition, several important questions remain unanswered at the late stage of elongation. First of all, we may also observe a torsion of the embryo, but if the muscle activity suggests a bending, this alone cannot explain alone a substantial torsion. Second, a small deviation of the muscle axis (*Moerman and Williams, 2006*) is responsible for a number of rotations, how to relate these rotations to the muscle activation (*Yang, 2017*). Since any measurement on a motile embryo at this scale is difficult, it is useful to explore the mechanism of late elongation theoretically. Furthermore, muscle contraction is crucial for both biological development and activities and has been extensively studied by researchers (*Tan and De Vita, 2015*; *De Vita et al., 2017*), but how it works at small scales remains to be understood.

Using a finite elasticity model and assuming that the embryonic body shape is cylindrical, we can evaluate the geometric bending deformation and the energy released during each muscle contraction on one side since after each contraction, the muscles relax, and then the muscles on the opposite side undergoes a new contraction. This cyclic process results in a tiny elongation of the cylinders along its axis of symmetry with each contraction. Each of these contractions correlates with the rotational movements of the embryo (*Yang, 2017*). By repeating these pairs of contractions more than two hundred times, a cumulative extension is achieved, but it must be reduced by friction mechanisms, also evaluated by the model. Furthermore, the mechanical model explains the existence of a torque acting on the position of the head or tail by the coupling of the muscle contraction with the orthoradial actomyosin forces. Finally, the small deviation between the muscles and the central axis experimentally detected (*Moerman and Williams, 2006*) induces cyclic rotations and possibly torsions leading to fluid viscous flow inside the egg. The quantification of all these processes allows the evaluation of

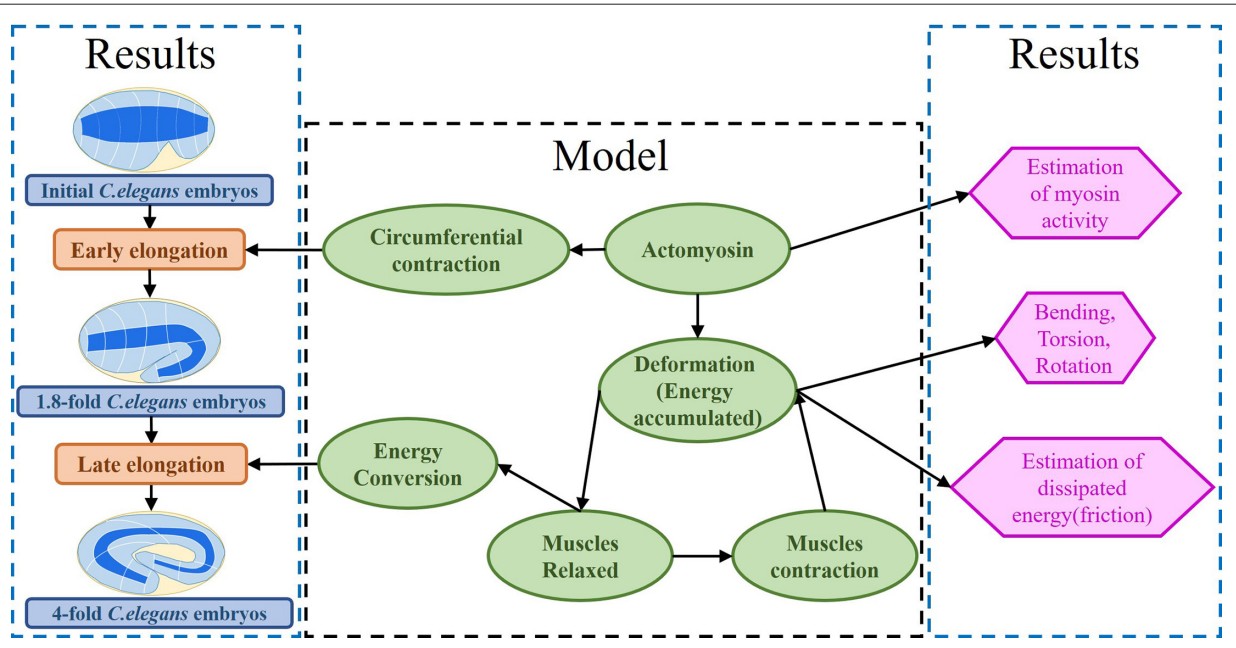

**Figure 2.** Architecture of the program. The program reflects the framework of the research. On the one hand, the proposed model explains the early and late elongation of the *C. elegans*, on the other hand, the early myosin activity is estimated, the deformations (bending, twisting, rotation) occurring in the late period are recovered, and the estimation of the energy dissipated during muscle activity is achieved.

**Table 1.** Adopted real size parameters of the *C. elegans* (**Vuong-Brender et al., 2017a**; **Ben Amar et al., 2018**).

|  | Initial | 1.8-fold |
|---|---|---|
| Radius | 11.1 µm | 8.2 µm |
| Length | 50 µm | 90 µm |

physical quantities of the embryo, such as the shear modulus of each component, the interstitial fluid viscosity, and the active forces exerted by the actomyosin network and the muscles, which are poorly known in the embryonic stages.

Our mechanical model accounts for the dynamic deformations induced by the internal stimuli of a layered soft cylinder, allowing us to make accurate quantitative predictions about the active networks of the embryo. Furthermore, our model accounts for the dissipation that occurs in the late period just before the embryo hatches. Our results are consistent with observations of actomyosin and muscle activity (**Vuong-Brender et al., 2017a**; **Ben Amar et al., 2018**; **Lardennois et al., 2019**). The architecture of our work is illustrated in *Figure 2*.

## Results

It is widely recognized that *C. elegans* is a well-established model organism in the field of developmental biology. However, it is less well known that its internal striated muscles share similarities with vertebrate skeletal muscles in terms of both function and structure (**Lesanpezeshki et al., 2021**). Since they are contractile, the role of the four axial muscles (as shown in *Figure 1*) in the final shaping of the embryo is almost counterintuitive. In this section, we aim to elucidate the action of these muscles when coupled to the actomyosin network, using a purely mechanical approach. To achieve a quantitative understanding, our method requires a detailed description of the deformation geometry of the embryo, where the body shape is represented by a fully heterogeneous cylinder.

### Geometric and strain deformations of the embryo

The early elongation of the *C. elegans* embryo has been previously studied, which is characterized by an inner cylinder surrounded by epithelial cells located in the cortical position (**Ben Amar et al., 2018**). The cortex is composed of three distinct cell types - the seam, dorsal, and ventral cells - that exhibit

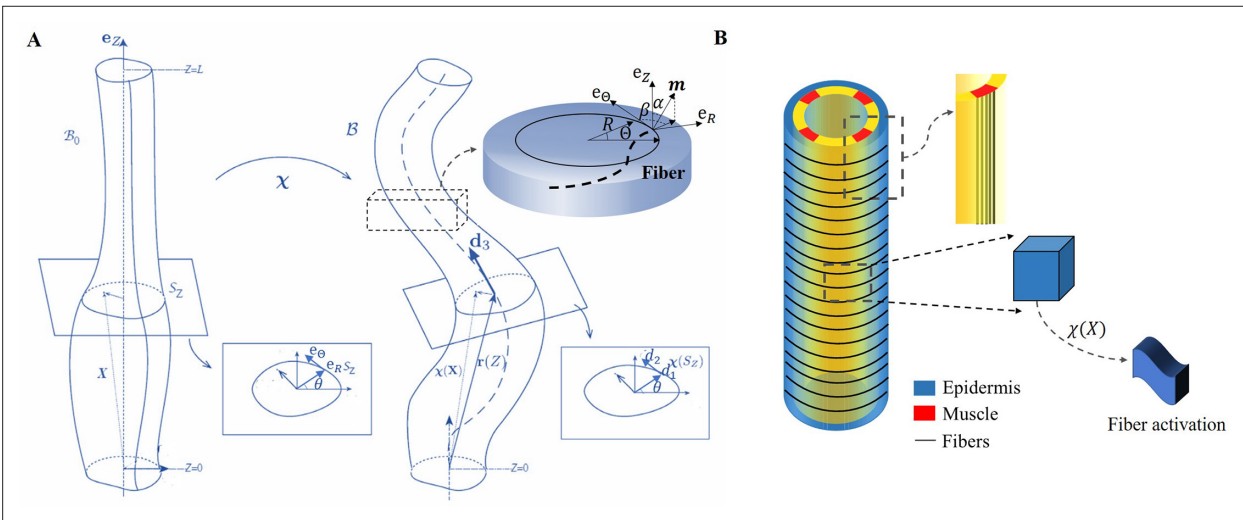

**Figure 3.** The model of *C. elegans* embryo. (**A**) Cylindrical structure in the reference configuration (left) with a vertical centerline and its deformation in the current configuration (right). The deformed configuration is fully parameterized by the centerline $\mathbf{r}(Z)$ (resulting from the distortion of the central axis) and the deformation of each cross-section. (**B**) Schematic representation of the body shape of the *C. elegans* embryo with the cortical epidermis and the four muscles. The fibers are embedded in the cortex. The blue part representing the epidermis shows the outer distribution of the actin organized in horizontal hoop bundles when the muscles are not activated. The yellow part includes the vertical red muscles, represented by axial fibers.

a unique cytoskeletal organization and actin network configuration. Among these, only the seam cells possess active myosin motors that function in the ortho-radial direction, allowing for the circumferential contraction, and thereby triggering the early elongation process. In this study, we simplify the cylindrical geometry of the body and treat the epidermis as a unified whole with effective activity localized along the circumference, supported by four muscles distributed beneath the epidermis. The muscles do not play a role in the early elongation phase and have, therefore, not been considered previously (*Vuong-Brender et al., 2017a*; *Ben Amar et al., 2018*). After this period of approximately a 1.8-fold increase in length, muscles parallel to the main axis and actin bundles organized along the circumference will cooperate to drive further elongation. *Table 1* provides the size parameters of *C. elegans* that will be used hereafter.

We focus on the total deformation of a full cylinder or a thin rod with a length $L$ greater than the radius $R$ and a central vertical axis along the $Z$ direction. Contrary to previous works (*Ciarletta et al., 2009*; *Ben Amar et al., 2018*), here we decide to simplify the geometrical aspect due to the mechanical complexity. The biological activity induced by the actomyosin network and muscles is represented by active strains, and the global shape is the result of the coupling between elastic and active strains, modulated by dissipation. Active strains are generated by non-mechanical processes (e.g. biochemical processes): myosin motors and striated muscles derive their energy from ATP hydrolysis, which is converted into mechanical fiber contractions. Due to the significant deformation observed, the central line is distorted and becomes a curve in three-dimensional space, represented by a vector $\boldsymbol{r}(Z)$, as depicted in *Figure 3A*. Along this curve, perpendicular planar sections of the embryo can be defined, and the deformation in each section can be quantified since the circular geometry is lost with the contractions (*Kaczmarski et al., 2022*). The geometric mapping is as follows:

$$\chi\left(\boldsymbol{X}\right) = \boldsymbol{r}\left(Z\right) + \sum_{i=1}^{3} \varepsilon a_i\left(\varepsilon R, \Theta, Z\right) \boldsymbol{d}_i\left(Z\right), \tag{1}$$

where the $a_i$ represents the deformation in each direction of the section with $a_i\left(0, \Theta, Z\right) = 0$ so that the $Z$-axis is mapped to the centerline $\boldsymbol{r}\left(Z\right)$. The small quantity $\varepsilon$ is the ratio between the radius $R$ and the length $L$ of the cylinder. The axial extension $\zeta$ is given by $\boldsymbol{r}'\left(Z\right) = \zeta \boldsymbol{d}_3$, where ' denotes the first derivative with respect to the material coordinate $Z$. From the director basis, the Darboux curvature vector reads: $\boldsymbol{u} = u_1 \boldsymbol{d}_1 + u_2 \boldsymbol{d}_2 + u_3 \boldsymbol{d}_3$, this vector gives the evolution of the director basis along the filamentary line as: $\boldsymbol{d}_i'\left(Z\right) = \zeta \boldsymbol{u} \times \boldsymbol{d}_i$, see *Figure 3A*. Based on the model proposed by *Kaczmarski et al., 2022*, we define the initial configuration $\mathcal{B}_0$ with material points $(R, \Theta, Z)$, and the mapping function $\chi\left(\boldsymbol{X}\right)$ connects the initial configuration $\mathcal{B}_0$ to the current configuration $\mathcal{B}$. The geometric deformation gradient is $\boldsymbol{F} = \text{Grad}\,\chi = \boldsymbol{F}_e \boldsymbol{G}$ (*Rodriguez et al., 1994*; *Nardinocchi and Teresi, 2007*), where $\boldsymbol{G}$ is the active strain generated by the actomyosin or the muscles, and $\boldsymbol{F}_e$ is the elastic strain tensor.

Because of the two stages through which the *C. elegans* elongates, we need to evaluate the influence of the *C. elegans* actin network during the early elongation before studying the deformation at the late stage. Thus, the deformation gradient can be decomposed into: $\boldsymbol{F} = \boldsymbol{F}_e \boldsymbol{G}_1 \boldsymbol{G}_0$ (*Goriely and Ben Amar, 2007*) where $\boldsymbol{G}_0$ refers to the pre-strain of the early period and $\boldsymbol{G}_1$ is the muscle supplementary active strain in the late period. Actin is distributed in a circular pattern in the outer epidermis, see *Figure 3B*, so the finite strain $\mathbf{G}_0$ is defined as $\boldsymbol{G}_0 = Diag\left(1, g_0(t), 1\right)$, where $0 < g_0 < 1$ is the time-dependent decreasing eigenvalue, operating in the actin zone, and is equal to unity in the other parts: $g_0(0) = 1$. In the case without pre-strain, $\boldsymbol{G}_0 = \boldsymbol{I}$. The deformation gradient follows the description of *Equation 1*: $\boldsymbol{F} = F_{ij}\boldsymbol{d}_i \otimes \boldsymbol{e}_j, where$ and $j \in \{R, \Theta, Z\}$.

Considering a filamentary structure *Figure 3(B)* with different fiber directions $\boldsymbol{m}$, these directions are specified by two angles $\alpha$ and $\beta$, as outlined in *Holzapfel, 2000*: $\boldsymbol{m} = \sin\alpha\sin\beta\,\boldsymbol{e}_{\boldsymbol{R}} + \sin\alpha\cos\beta\,\boldsymbol{e}_{\boldsymbol{\Theta}} + \cos\alpha\,\boldsymbol{e}_{\boldsymbol{Z}}, \ \alpha, \beta \in \left[-\pi/2, \pi/2\right]$. For muscle fibers, $\alpha_m = 0$ and $\beta_m = 0$, while for hoop fibers in the actin network, $\alpha_a = \pi/2$ and $\beta_a = 0$. When the muscles are activated and bend the embryo, the actin fibers are tilted with a tilt $\alpha$, such that $-\pi/2 < \alpha_a < \pi/2, \ \alpha_a \neq 0$, and $\beta_a = 0$. Each active strain is represented by a tensor $\boldsymbol{G}_i = \boldsymbol{I} + \varepsilon g_i \boldsymbol{m}_i \otimes \boldsymbol{m}_i$, where $g_i$ represents the activity ($g_m$ for muscles and $g_a$ for actomyosin), and since both are contractile, their incremental activities are negative. Additional calculation details are provided in section Methods and materials.

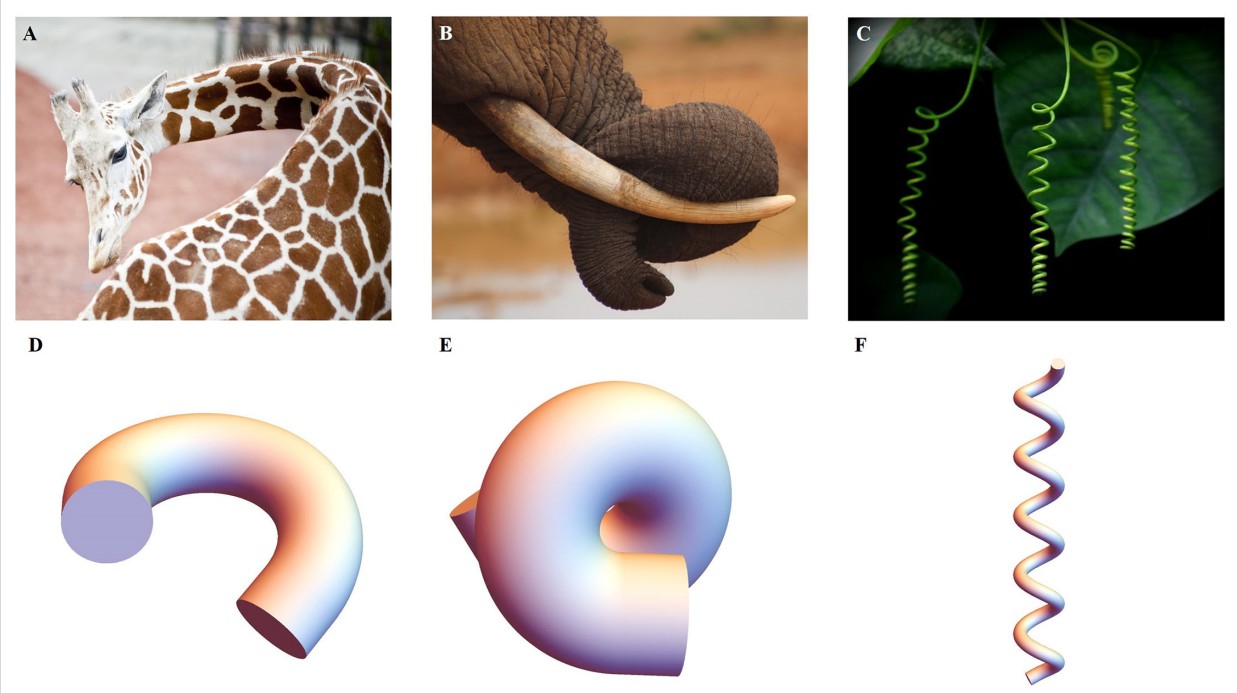

**Figure 4.** Extended application of the model. (**A**) Bending of a giraffe's neck. (**B**) Torsion of an elephant's trunk. (**B**) Twisting of a plant vine. (**D**) to (**F**) Deformation configuration under different activations obtained by our simulations for bending and torsion of large rods, twisting, and torsion of thin rods.

To relate the deformations to the active forces induced by the muscles and the actomyosin network, we assume that the embryo can be represented by the simplest nonlinear hyperelastic model, called Neo-Hookean, with a strain energy density given by $W\left(\boldsymbol{F}_e\right) = \frac{\mu}{2}\left(tr\left(\boldsymbol{F}_e\boldsymbol{F}_e^T\right) - 3\right)$. In cylindrical coordinates, the total energy of the system, and the associated energy density is:

$$\mathcal{W} = \varepsilon^2 \int_0^L \mathrm{d}Z \int_s V(\mathbf{F_e}, \mathbf{G}) R\, \mathrm{d}R\, \mathrm{d}\Theta,$$
$$V = W(\mathbf{F_e}) - p(J - 1). \tag{2}$$

where $J = \det \mathbf{F}_e$, $p$ is a Lagrange multiplier that ensures the incompressibility of the sample, a physical property assumed in living matter. If the cylinder contains several layers with different shear modulus µ and different active strains, the integral over $S$ covers each layer. To minimize the energy over each section for a given active force, we take advantage of the small value of $\varepsilon$ and expand the inner variables $a$, $p$, and the potential $V$ to obtain the associated strain-energy density. Since the Euler-Lagrange equations and the boundary conditions are satisfied at each order, we can obtain solutions for the elastic strains at zero order $\mathbf{a^{(0)}}$ and at first order $\mathbf{a^{(1)}}$. Finally, these solutions which will represent a combination of bending and torsional deformation, will last for the duration of a muscle contraction as the actomyosin continues its contractile activity. The evaluation of the elastic energy under actomyosin muscle activity of order $\varepsilon^4$ can be compared with the equivalent energy of an extensible elastic rod and any fundamental deformation will be identified (*Kirchhoff and Hensel, 1883*; *Mielke, 1988*; *Mora and Muller, 2003*; *Mora and Müller, 2004*; *Moulton et al., 2020a*). The typical quantities of interest are the curvature achieved during a bending event or a torsion as well as the total elongation $\zeta$. In the *C. elegans* embryonic system, these quantities result from the competition between the active strains due to muscles and the myosin motors and the elasticity of the body. Similar active matter can be found in biological systems, from animals to plants, as illustrated in *Figure 4A–C*, they have a structure that generates internal stress/strain for the growth or daily activities. By combining anatomy and measurement techniques, we can transform the mechanics of the body under study into a soft sample subjected to localized internal active stresses or localized internal active strains and then deduce its overall deformation mechanisms, some examples are presented in *Figure 4D–F*.

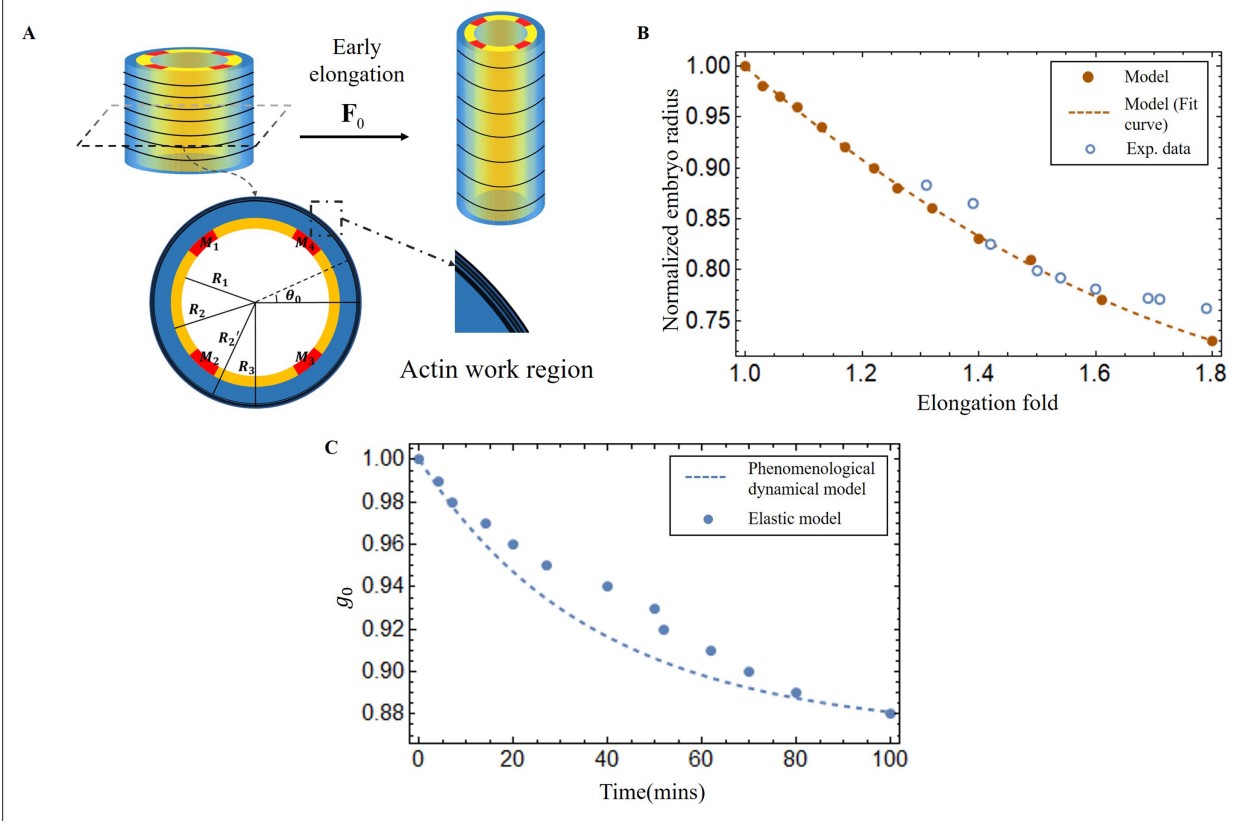

**Figure 5.** Early elongation model of *C. elegans* embryo and validation images of the model results. (**A**) Schematic of early elongation and the cross-section of *C. elegans*. In the cross-section, the black circular part is the actin region ($R_2' < R < R_3$, with shear modulus $\mu_a$), the blue part is the epidermis layer ($R_2 < R < R_2'$ with shear modulus $\mu_e$). The central or inner part ($0 < R < R_2$) in white has shear modulus $\mu_i$, except for the four muscles, the shear modulus of the four muscles $\mu_m$ is much larger than the inner part. All detailed data are given in *Appendix 1—table 1* (**B**) Predictions of normalized embryo radius evolution during early elongation by the pre-strain model compared to experimental data from *Vuong-Brender et al., 2017a*. For the model, see *Equation 20* in Appendix 2. (**C**) Blue dots: extraction of the parameter $g_0(t)$ from *Equation 30* and *Equation 32* in Appendix 2. Blue dash line, see *Equation 3*.

## The early elongation induced by the actomyosin

Experimental measurements during the early elongation stage reveal the embryonic diameter and the active or passive stresses, estimated by the opening of fractures realized in the body by laser ablation (*Vuong-Brender et al., 2017a*; *Ben Amar et al., 2018*). These quantities vary with the elongation. While previous studies have extensively investigated this initial stage, we have chosen to revisit it in the context of our geometry, with the goal of achieving complete control over our structural modeling, which includes the geometry, shear modulus, and activity of the actomyosin prior to muscle contraction. At the beginning of the second stage, the embryo experiences a pre-strain due to the early elongation, which we describe in our model as $\mathbf{G}_0$. The geometry and mechanical information are depicted in *Figure 5A*.

Here, we must first accurately determine $\mathbf{G}_0$ by analyzing the experimental data (*Vuong-Brender et al., 2017a*; *Ben Amar et al., 2018*). The cylinder can be divided into three distinct sections: the outer layer is the actin cortex, the thin ring where actin bundles concentrate and work, located within a radius range between $R_2'$ and $R_3$ (as shown in *Figure 5A*); the middle layer ($R_2 < R < R_2'$), which is the epidermis but without actin; and the inner part ($0 < R < R_2$), where the muscle is located, along with some internal organs, tissues, and fluids. So, we treat the outer and middle layers as incompressible, but the inner part as a compressible material, except for the muscles. The initial deformation gradient: $\mathbf{F}_0 = \text{Diag}\,(r'(R), r(R)/R, \lambda)$, and $\mathbf{G}_0 = \text{Diag}\,(1, g_0, 1)$ with $0 < g_0 < 1$ in the actin layer, but with $g_0 = 1$ in the part without actin. $\mathbf{G}_0$ represents the circumferential strain exerted by actin during the early elongation and is a slowly varying function of time. By applying the principles of radius continuity, radial stress continuity, and incorporating the zero traction condition on the face of the cylinder,

we can determine that $g_0 = 0.88$ when the elongation $\lambda = 1.8$, at the end of the early elongation. As illustrated in *Figure 5B*, the results of our model and the experimental data are in good agreement, demonstrating the consistency of the geometric and elastic modeling together with the choice of a pre-strain represented by $\mathbf{G_0}$ which gives a good prediction of the early elongation. More details can be found in the Appendix 3.

From the first stage elongation represented by the blue dots and the blue dashed line in *Figure 5C* we can extract the time evolution of the contractile pre-strain $g_0(t_i)$ derived from our elastic model. To explain $g_0(t)$ quantitatively, we propose a phenomenological dynamical approach for the population of active myosin motors. This equation takes into account the competition between the recruitment of new myosin proteins from the epidermal cytoskeleton, which is necessary to elongate the embryo, and the detachment of these myosins from the actin cables, which is damped by the compressive radial stress. It reads:

$$\frac{dX_g(t)}{dt} = \left( p_1 - p_2 X_g(t) e^{-p_3 X_g(t)} \right) \frac{t_v}{t_p} \tag{3}$$

where $X_g = 1 - g_0(t)$, $p_1$ is the ratio of the freely available myosin population to the attached ones divided by the time of recruitment (given in minutes), while $p_2$ is the inverse of the debonding time of the myosin motors from the cable: $p_2 = 6$ min$^{-1}$. The debonding time increases (or decreases) when the actin cable is under radial compressive (or tensile) stress, see Appendix 3, *Equation 35*. $\tau_v$ is the

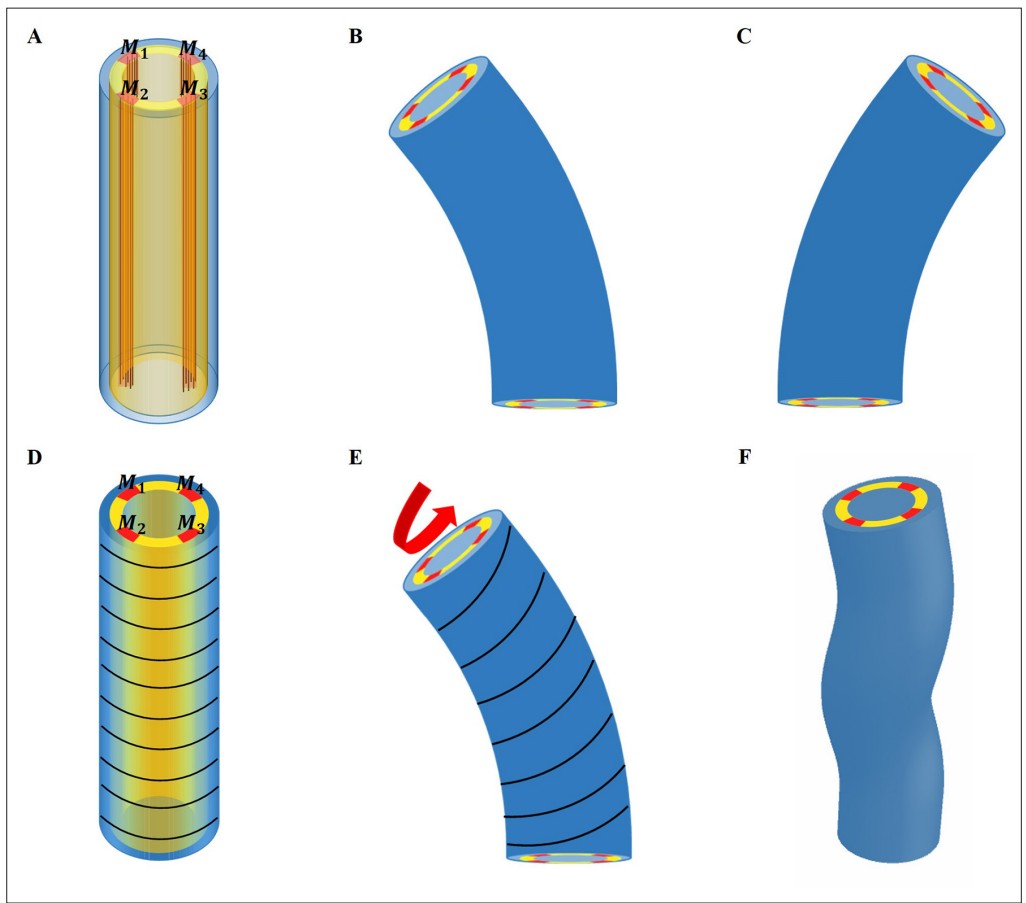

**Figure 6.** Schematic representation of the model deformation under activation. (**A**) Schematic diagram of *C. elegans* muscle fibers and its cross-section, and it does not show the actin fibers. There are four muscle bands that exist in the yellow layer. However, the yellow region is not an actual tissue layer and it is only used to define the position of the muscles. (**B**) Deformation diagram, left side muscles $M_1$ and $M_2$ are activated. (**C**) Deformation diagram, right side muscles $M_3$ and $M_4$ are activated. (**D**) Schematic diagram of *C. elegans* actin fibers and cross-section. (**E**) Once the muscle is activated, the actin fiber orientation changes from the 'loop' to the 'slope,' resulting in torque. (**F**) Schematic diagram of torsional and bending deformation.

viscoelastic time estimated from laser ablation fracture operated in the epidermis (*Vuong-Brender et al., 2017b*): $t_v = 6s$ and $\tau_p$ is the time required to activate the myosin motors $t_p = 1200s$ (*Howard, and Clark,, 2002*. This equation is similar to the model derived by *Serra et al., 2021*) for the viscous stress that occurs during gastrulation.

Note that only $p_1$ and $p_3$ need be obtained by comparing $g_0(t)$ deduced from *Equation 3* with the values derived from our elastic model. The result of *Equation 3* with $p_1 = 0.6$ and $p_3 = 0.75$ are shown in *Figure 5C* with a rather good agreement.

## Shape of the embryo under muscles and actomyosin contraction

The experimental regulation of muscle contraction in *C. elegans* (*Yang, 2017*) suggests a cyclic process in which two muscles on one side of the embryo contract quite at the same time and then stop, while on the opposite side, the two muscles begin to contract, but with a delay. Let us first consider that only the muscles are active (see the schematic *Figure 6A* for the structure, then (B) and (C) for the bending). In this case, due to the geometry, bending to the left occurs when the left muscles are activated and then to the right for the symmetrical right muscles.

To be more quantitative, we assume that the left side muscles are activated during a short period with an active constant strain value $g_m$ in the region $M_1$ and $M_2$, as shown in *Figure 6A, B*; if the muscles are perfectly vertical, $\alpha_m = \beta_m = 0$ in the initial configuration. In fact, the two muscles on the same side are not rigorously in phase, and one may have a small delay. For simplicity, we assume that they contract simultaneously. During the entire initial period when the muscles are not activated, the actin fibers are distributed in a horizontal loop on the outer surface of the epidermis, but as soon as the muscle starts to contract, the actomyosin network will be reoriented (*Lardennois et al., 2019*). The fibers are then distributed in an oblique pattern, which eventually causes the twisting of the embryo, see the schematic diagram in *Figure 6D–F*. When this region is activated with a constant strain value, $g_a$, the angle of the actin fibers will change according to the amplitude of the bending caused by the muscle contraction. In this situation, the angle of the actin fibers may change from $\alpha_a \in \left[0, \frac{\pi}{2}\right]$, but $\beta_a$ is not changed and $\beta_a = 0$.

Throughout the entire process, the muscle and actomyosin activities are assumed to operate simultaneously. Our modeling allows us to evaluate the bending and torsion generated independently by muscles and actin bundles, culminating in a complete deformation under coupling. Furthermore, the angle of the actomyosin fibers varies during muscle contraction. We maintain a constant activation of the actomyosin network and gradually increase the muscle activation.

As a result, the structure will be bent, causing a change in the angle of the actin fibers, ultimately yielding a deformation map that is presented in *Figure 7A–C*. As myosin activation increased, we observed a consistent torsional deformation (*Figure 7E*) that agrees with the patterns seen in the video (*Figure 7D*). However, significant torsional deformations are not always present. In fact, other sources can induce torsion as a lack of symmetry of the muscle axis. We will now discuss torsion due to muscle contraction and the angle of deviation from the axis. In *Figure 7F, G*, we demonstrate that the curvature provided by the model increases with muscle activation and that the torsion is not simply related to the activation amplitude, as it also depends on the value of the angle $\alpha_a$, reaching a maximum at about $\pi/4$. Detailed calculations are given in section Methods and materials and Appendix 4.

## Energy transformation and elongation

During the late elongation process, the four internal muscle bands contract cyclically in pairs (*Yang, 2017*; *Williams and Waterston, 1994*). Each contraction of a pair increases the energy of the system under investigation, which is then rapidly released to the body. This energy exchange causes the torsional bending energy to be converted into elongation energy, leading to an increase in length during the relaxation phase, as shown in *Figure 1* of the Appendix 5. With all the deformations obtained, *Equation 2* can be used to calculate the accumulated energy $W_c$ produced by both the muscle and the actomyosin activities during a contraction. Subsequently, when the muscles on one side relax, the worm body returns to its original shape but with a tiny elongation corresponding to the transferred elastic energy. In this new state, the actin network assumes a 'loop' configuration with a strain of $\varepsilon g_{a1}$ once relaxation is complete. If all the accumulated energy from the bending-torsion deformation is used to elongate the worm body, the accumulated energy $W_c$ and the energy $W_r$ after

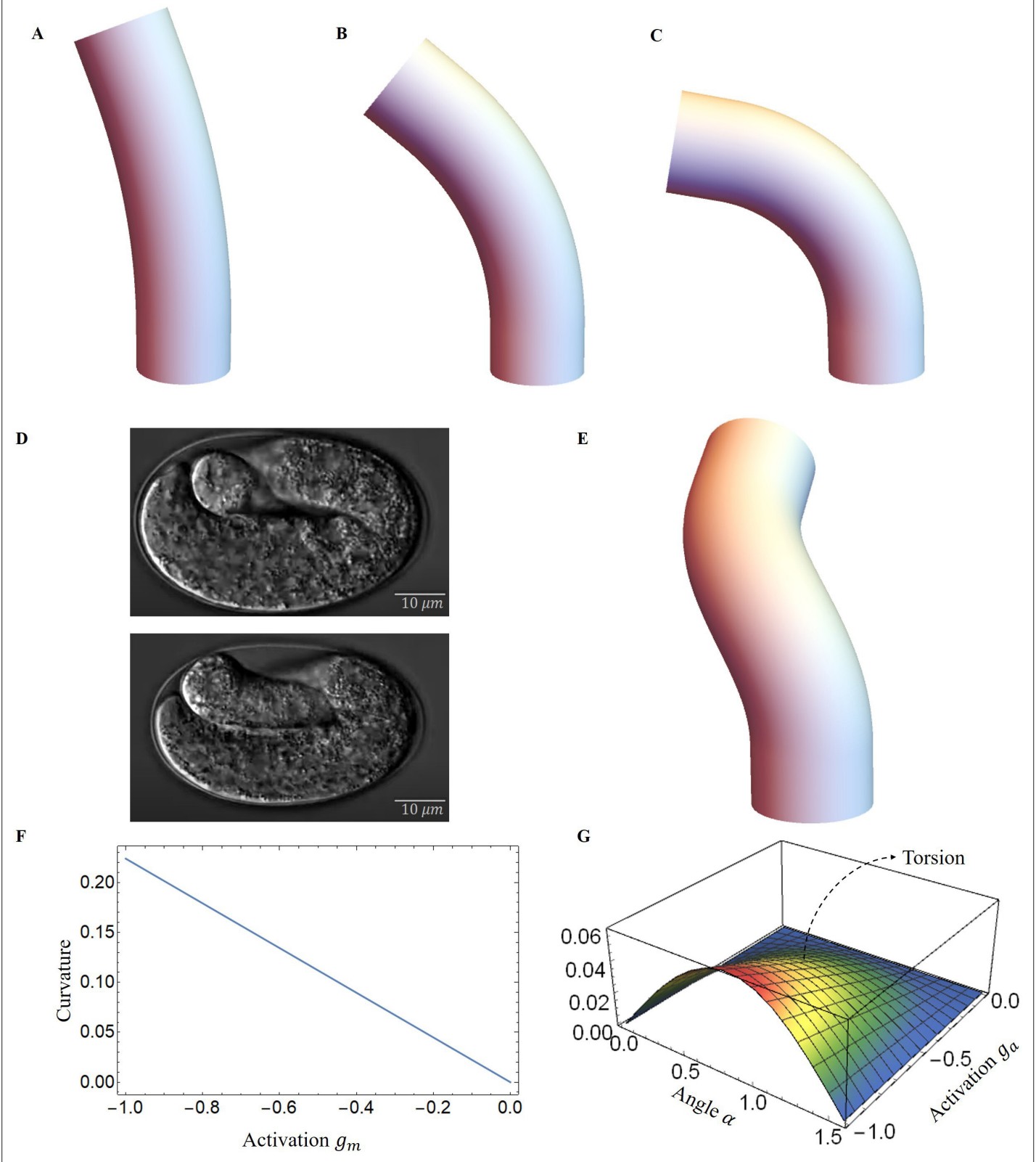

**Figure 7.** Deformed image of the model with activation changes. Deformed configurations for different activation for muscles, (**A**) $g_m = -0.02$, $\alpha_a = \pi/3$, $g_a = -0.01$. (**B**) $g_m = -0.05$, $\alpha_a = \pi/4$, $g_a = -0.01$. (**C**) $g_m = -0.08$, $\alpha_a = \pi/6$, $g_a = -0.01$. (**D**) The graphs were captured from the Hymanlab, and the website: https://www.youtube.com/watch?v=M2ApXHhYbaw. The movie was acquired at a temperature of 20°C using DIC optics. (**E**) $g_m = -0.1$, $\alpha_a = \pi/4$, $g_a = -0.7$. (**F**) Curvature is plotted as a function of muscle activation. (**G**) Torsion is plotted as a function of the actin activation and angle of actin fibers.

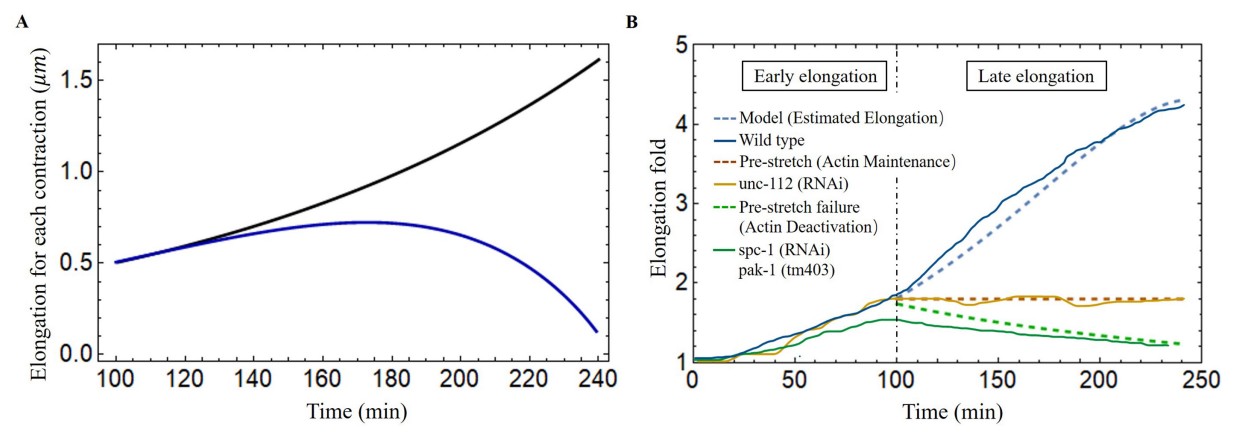

**Figure 8.** Late elongation predicted result for the *C. elegans* model. (**A**) The elongation for each contraction varies with time. Black line: all energy converted to the elongation, blue line: partial energy converted to the elongation. The activation: $g_m = -0.15$, $g_a = -0.01$. (**B**) The model predicted results agree well with the experimental data of wild-type and different mutant *C. elegans* embryos (**Lardennois et al., 2019**). Activation of the wild-type model (blue dashed line): $g_m = -0.15$, $g_a = -0.01$. The activation of unc-112(RNAi) (brown dashed line): $g_m = 0$, $g_a = 0$. In the case of pre-stretch failure (green dashed line), $\lambda$ decreases from 1.8.

muscle relaxation are equivalent. The activation of actin fibers $g_{a1}$ after muscle relaxation can be calculated and determined by our model.

Once the first phase is well characterized, we can quantify the total energy resulting from both muscle and actomyosin after each contraction. We then used our model to predict elongation in wild-type *C. elegans*, unc-112(RNAi) mutant and spc-1(RNAi) pak-1(tm403) mutant and further compared the results with experimental observations. The wild-type length increases from about 90 to 210 during the muscle-activated phase (**Lardennois et al., 2019**), the phase lasts about 140 min, and the average time interval between two contractions is about 40 s (**Yang, 2017**), so the number of contractions can be estimated to be about 210 times. Unfortunately, due to the difficulty of performing quantitative experiments on an embryo that is constantly moving in the egg shell between bending, torsion, and even rotations around its central axis, one can hypothesize this scenario: at each step $i$ between the state $A_{i,0}$ to $A_{i,1}$ and then $A_{i,2}$, the whole mechanical muscle-myosin energy is transferred to the elongated step $A_{i,2}$, after resulting in a small $\delta\zeta_i$.

Considering the experimental results shown in **Figure 8B**, we determine the optimal values for the activation parameters: $g_m = -0.15$ and $g_a = -0.01$ assuming that all the energy accumulated during the muscle activation is transferred to the elongation process ($W_r = W_c$). The elementary elongation $\delta\zeta_i$ is gradually increased over time, which is shown as the black line in **Figure 8A**. At the beginning, $\delta\zeta_i$ is about $0.5\mu m$, but at the end of this process, $\delta\zeta_i$ is about $1.5\mu m$, indicating that the worm will grow up to $290\mu m$. The result is significantly larger than our actual size $210\mu m$. As the elongation progresses, we assume that there is an energy transfer between bending-torsion-contraction and elongation but it may not be fully effective, meaning that a significant part of the energy is lost. From the experimental data, we estimate that the energy loss gradually increases, from full conversion at the beginning to only 40% of the accumulated energy used for elongation at the end of the process ($W_r = 0.4W_c$). For $\delta\zeta_i$ it induces a first increase and then a decrease, which is shown as the blue line in **Figure 8A** and is responsible for the slowdown around 200 min. This possibility, which may be not the only possible one, leads to the estimated elongation being in good agreement with the experimental data (see the blue-dashed curve in **Figure 8B**). Indeed it is possible that the *C. elegans* elongation requires other transformations that cost energy. As the embryo gradually elongates, energy dissipation, and the biomechanical energy required to reorganize the actin bundles may be two factors contributing to the increased energy loss underlying the hypothesis.

In addition, it has been reported in **Norman and Moerman, 2002** that the knockdown of unc-112(RNAi), known the to affect muscle contractions, leads to the arrest of elongation of the embryos at the twofold stage, indicating that the muscles have no activation, $g_m = 0$ in our model, and no accumulated energy can be converted into elongation. Another mutation affecting the embryos consisting of mutant cells with pak-1(tm403), known to regulate the activity of myosin motors

leads to a retraction of the embryo, so that the pre-stretch caused by myosin is not maintained and decreases. The results are fully consistent with a number of experimental observations and are shown in *Figure 8B*.

## Embryo rotations and dissipation

The main manifestation of the muscle activity, independent of the elongation, is probably the constant rotation of the embryo despite its confinement in the egg. This can be explained by a small angular deviation of the muscle sarcomeres from the central axis due to their attachment to the inner boundary of the cell epidermis, the so-called 'dense bodies' (*Moerman and Williams, 2006*). Since they cross the horizontal plane at about ±45° and the deviation $\beta_m$ from the anterior-posterior axis is estimated to be about 6° each active muscle on the left (or on the right) contributes to the torque via a geometric factor of about $a_g = \sin(6\pi/180)\cos(\pi/4)$. Then a simple estimate of muscle activity in terms of torque is $\Lambda_m \sim \mu_m \pi R^3 s_m p_m a_g (\varepsilon g_m)$ where $s_m$ is the area of the left or right muscle pair compared to the section of the cylinder: $s_m = 0.025$ and $p_m$ is the distance of the muscles from the central axis of the embryo: $p_m = 0.75$ while $g_m = 0.15$ according to the elongation analysis. So the muscles on one side contribute to a torque $\Lambda_m$ along the symmetry axis, given by $\Lambda_m = 4.657 \mu_m \pi R^3 \cdot 10^{-5}$.

Let us now consider the dissipative torque, assuming that the rotational dynamics are stopped by friction after a bending event. Two cases can be considered: either the dissipation comes from the viscous flow or from the rubbing of the embryo when it folds. The fluid dissipation results from the rotations in the interstitial fluid inside and along the eggshell as the anterior-posterior axis remains parallel to the eggshell (*Bhatnagar et al., 2023*). The interstitial fluid, of viscosity $\eta$ contains a significant amount of sugars and other molecules necessary for embryo survival and that is more viscous than water (*Soesanto and Williams, 1981*; *Chadwick, 1985*; *Bouchard et al., 2007*; *Telis et al., 2007*; *Hidayanto et al., 2010*; *Labouesse, 2023*). However, values for sucrose or sorbitol at the concentration of 1 mole/liter indicate a viscosity on the order of the viscosity of water, which is 1 mPas. For example at 0.9 mole/liter and temperature of 20°, an aqueous solution of sorbitol has a viscosity of 1.6 mPas, which can be extrapolated to $\eta = 1.9$ mPas at 1.2 mole/liter (*Jiang et al., 2013*). The estimate given by an embryo located in the middle of the egg shell, gives a weaker viscous torque once evaluated by $\Lambda_v = 4\pi\eta\Omega_e L R^2 \left( R_{egg}^2/(R_{egg}^2 - R^2) \right)$ according to a classical result reported by *Landau and Lifshitz, 1971*; *Landau and Lifshitz, 2013*.

It should be noted that this estimation assumes that the two cylinders: the egg shell and the embryo have the same axis of symmetry and concerns the beginning of the muscle activity where the radius is about $8.2\,\mu m$, the length is $90\,\mu m$, the radius of the shell is about $R_{egg} = 15\mu m$ and the length $L_{egg} = 54\mu m$, see *Figure 7D*. The angular velocity $\Omega_e$ is more difficult to evaluate but it is about 90° per 2 s deduced from videos. As the embryo approaches the eggshell, the friction increases, and two eccentric cylinders of different radii must be considered with the two symmetry axes separated by a distance $d$. The hydrodynamic study in this case is far from being trivial, and seems to have been first initiated by *Zhukoski, 1887*, who proposed the use of bipolar coordinates for the mathematical treatment. Many subsequent papers were published after, using barious simplified assumptions, and the study was completely revisited by *Ballal and Rivlin, 1976*. Here, we focus on the limit of a small gap $\delta$ between the rotating body shape and the egg and by considering an asymptotic analysis at small $\delta$ of the general result derived in *Ballal and Rivlin, 1976*. Thus, the viscous torque reads $\tilde{\Lambda}_v = 2\sqrt{2}\pi\eta\Omega_e L_c R_{egg}^2 \sqrt{R_{egg}/\delta}\sqrt{(R_{egg} - d)/d}$, where $L_c$ is the zone of contact with the egg and $d = R_{egg} - \delta - R$. This approach is an approximation since the embryo is more toroidal than cylindrical (*Yang, 2017*), but the evaluation of the dissipation is satisfactory for $\delta = 0.5\mu m$, $\Omega_e \sim \pi/4\,s^{-1}$ and $\mu = 10^5 Pa$. Going back to the first model of dissipation with the same data, the ratio between the dissipative viscous torque and the active one gives: $\Lambda_v/\Lambda_m = 0.02$, which is obviously unsatisfactory. Finally, the dissipative energy $\mathcal{E}_{diss}$ during one bending event leading to an angle of $\pi/2$ is $\mathcal{E}_{diss} = 1/2\Lambda_m \times (\pi/2)^2$ which represents 4% of the muscle elastic energy during the bending so at the beginning of the muscle activity (Appendix 5, *Equation 45*), the dissipation is present but is negligible. At the very end of the process, this ratio becomes 60% but as already mentioned, our estimate for the dissipation becomes very approximate, increases a lot due to the embryo confinement, and does not include the numerous biochemical steps necessary to reorganize the active network: actomyosin and muscles.

## Discussion

Since the discovery of the pre-embryonic muscle activity in the *C. elegans* embryo, it has become critical to explain the role of mechanical forces generated by muscle contraction on the behavioral and functional aspects of the epidermis. We provide a mechanical model in which the *C. elegans* is simplified as a cylinder, and the muscle bands and actin that drive its elongation are modeled as active structures in a realistic position. We determine the fiber orientation from experimental observations and then calculate the deformation by tensorial analysis using the strains generated by the active components. Although a special focus is given to late elongation, its quantitative treatment cannot avoid the influence of the first stage of elongation due to the actomyosin network, which is responsible for a pre-strain of the embryo. In a finite elasticity formalism, the deformations induced by the muscles in a second step are coupled to the level of strains of the initial elongation period. For this, we must revisit the theory of the actomyosin contraction and previous results (*Ciarletta et al., 2009*; *Vuong-Brender et al., 2017a*; *Ben Amar et al., 2018*) to unify the complete treatment. In particular, a model for the recruitment of active myosin motors under forcing is presented that recovers the experimental results of the first elongation phase.

The elongation process of *C. elegans* during the late period is much more complex than the early elongation stage which is caused by actin contraction alone. During late elongation, the worm is deformed by the combined action of muscle and actomyosin, resulting in an energy-accumulating process. Bending deformation is a phenomenon resulting from unilateral muscle contraction, and during the late elongation, significant torsional deformation is observed, indicating that the bending process induces a reorientation of the actin fibers. It should be noted that the embryo always rotates in parallel to the muscle activity, which makes any experimental measurements difficult. However, our model can predict that if the muscles are not perfectly vertical, a torque is generated that causes rotation and eventually torsion. The accumulated energy is then partially converted into energy for the ongoing action of actin, allowing the embryo to elongate when the muscle relaxes. Both sides of the *C. elegans* muscles contract in a sequential cycle, repeating the energy conversion process, and eventually completing the elongation process. However, the energy exchange between bending and elongation is limited, among other factors, by the viscous dissipation induced by rotation, which is also evaluated in this study. The necessary reorganization of the active networks (actomyosin and muscles) due to this enormous shape change of the embryo is not investigated here in detail. Parallel to the elongation, the cuticle is built around the body (*Page and Johnstone, 2007*). This very thin and stiff membrane provides protection and locomotion after hatching. Obviously, these processes will interfere with muscle activity. These two aspects, which intervene in the final stage of the worm's confinement, play a very important role at the interface between genetics, biochemistry, and mechanics.

Finally, the framework presented here not only provides a theoretical explanation for embryonic elongation in *C. elegans*, but can also be used to model other biological behaviors, such as plant tropism (*Moulton et al., 2020b*) and elephant trunk elongation (*Schulz et al., 2022a*; *Schulz et al., 2022b*) and bending. Our ideas could potentially be used in the emerging field of soft robotics, such as octopus leg-inspired robots (*Kang et al., 2012*; *Nakajima et al., 2013*; *Calisti et al., 2015*), which are soft and their deformation is induced by muscle activation. We can reliably predict the deformation by knowing the position of activation and the magnitude of the forces in the model. In addition, residual stresses can be incorporated into our model to achieve design goals.

## Materials and methods

The model has been presented in a series of articles by *Goriely and Tabor, 2013*; *Moulton et al., 2013*; *Moulton et al., 2020b*; *Kaczmarski et al., 2022*; *Goriely et al., 2023*. As one can imagine, this is far from trivial, and most of the literature on the subject concerns either full cylinders or cylindrical shells in torsion around the axis of symmetry, and the case of bending is far from trivial. However, the geometry of the muscles in *C. elegans* automatically leads to a bending process that cannot be discarded. We take advantage of these previous works and apply the methodology to our model.

### The deformation gradient

The finite strain $\mathbf{G}_0$ maps the initial stress-free state $\mathcal{B}_0$ to a state $\mathcal{B}_1$, which describes the early elongation. We then impose an incremental strain field $\mathbf{G}_1$ that maps the state $\mathcal{B}_1$ to the state $\mathcal{B}_2$, which

represents late elongation. So, the deformation gradient can be expressed as: $\mathbf{F} = \mathbf{F}_e\mathbf{G}_1\mathbf{G}_0$ (*Goriely and Ben Amar, 2007*). The deformation gradient $\mathbf{F} = F_{ij}\mathbf{d}_i \otimes \mathbf{e}_j$, and $j$ labels the reference coordinates $\{R, \Theta, Z\}$ finally reads:

$$\boldsymbol{F} = \begin{bmatrix} a_{1R} & \frac{1}{R}a_{1\Theta} & \lambda\varepsilon\left(1 + \varepsilon\xi\right)\left(u_2a_3 - u_3a_1\sin a_2\right) \\ a_1a_{2R} & \frac{a_1}{R}a_{2\Theta} & \lambda\varepsilon\left(1 + \varepsilon\xi\right)\left(u_3a_1\cos a_2 - u_1a_3\right) \\ a_{3R} & \frac{1}{R}a_{3\Theta} & \lambda\left(1 + \varepsilon\xi\right)\left(1 + \varepsilon\left(u_1a_1\sin a_2 - u_2a_1\cos a_2\right)\right) \end{bmatrix}, \tag{4}$$

where $\lambda$ is the axial extension due to the pre-strained.

We define $\mathbf{G}_0 = Diag\left(1, 1 + \varepsilon c, 1\right)$, since $g_0 = 0.88$ can be determined by the early elongation, we write it in the form of $1 + \varepsilon c$ to simplify the calculation, $c \approx -0.6$ ($\varepsilon \approx 0.2$).

Consider fibers oriented along the unit vector $\mathbf{m} = (\sin\alpha\sin\beta, \sin\alpha\cos\beta, \cos\alpha)$ for the incremental strain $\mathbf{G_1}$, where $\alpha$ and $\beta$ characterize the angles with $\mathbf{e_z}$ and $\mathbf{e_\theta}$ and can be found in *Figure 3A*. The active filamentary tensor in cylindrical coordinates is then $\mathbf{G} = \mathbf{G}_0\left(\mathbf{I} + \varepsilon g\,\mathbf{m} \otimes \mathbf{m}\right)$, see *Holzapfel, 2000* which reads:

$$\boldsymbol{G} = \boldsymbol{G_0} + \varepsilon g\boldsymbol{G_0}\begin{bmatrix} \sin^2\alpha\sin^2\beta & \sin^2\alpha\sin\beta\cos\beta & \sin\alpha\cos\alpha\sin\beta \\ \sin^2\alpha\sin\beta\cos\beta & \sin^2\alpha\cos^2\beta & \sin\alpha\cos\alpha\cos\beta \\ \sin\alpha\cos\alpha\sin\beta & \sin\alpha\cos\alpha\cos\beta & \cos^2\alpha \end{bmatrix}. \tag{5}$$

## The energy function

The associated strain-energy density can be obtained by expanding the inner variables $a_i$ (see *Equation 1*), $p_i$, and the potential $V$ (see *Equation 2*) as:

$$\boldsymbol{a} = \boldsymbol{a}^{(0)} + \varepsilon\boldsymbol{a}^{(1)} + \varepsilon^2\boldsymbol{a}^{(2)} + \mathcal{O}\left(\varepsilon^3\right), \tag{6}$$

$$p = p^{(0)} + \varepsilon p^{(1)} + \varepsilon^2 p^{(2)} + O\left(\varepsilon^3\right), \tag{7}$$

$$V(\boldsymbol{F}_e, \boldsymbol{G}) = V_0 + \varepsilon V_1 + \varepsilon^2 V_2 + O\left(\varepsilon^3\right). \tag{8}$$

For each cross-section, the associated Euler-Lagrange equations take the following form:

$$\frac{\partial}{\partial R}\frac{\partial V_iR}{\partial a_{jR}^{(k)}} + \frac{1}{R}\frac{\partial V_iR}{\partial a_{jR}^{(k)}} + \frac{\partial}{\partial\Theta}\frac{\partial V_iR}{\partial a_{j\Theta}^{(k)}} - \frac{\partial V_iR}{\partial a_j^{(k)}} = 0, \tag{9}$$

where $j = 1, 2, 3, k = 0, 1$, in association with boundary conditions at each order $k = 0, 1$ on the outer radius $R_i$:

$$\left.\frac{\partial V_iR}{\partial a_{jR}^{(k)}}\right|_{R=R_i} = 0, j = 1, 2, 3. \tag{10}$$

We solve these equations order by order, taking incompressibility into account. At the lowest order, the incompressibility imposes the deformation: $\mathbf{a}^{(0)} = \left(r\left(R\right), \Theta, 0\right)$ and the Euler-Lagrange equation gives the Lagrange parameter $p^{(0)} = P_0(R)$ which finally read:

$$r' = \frac{R}{\lambda r}, \; P_0' = 0, \tag{11}$$

and the boundary condition $\sigma_{rr}(R = 1) = 0$ gives the value of $P_0 = 1/\lambda$. At $\mathcal{O}\left(\varepsilon\right)$, the Euler-Lagrange equations are again automatically satisfied, and at $\mathcal{O}\left(\varepsilon^2\right)$, the crucial question is to get the correct expression for $a_i^{(1)}$ and $p^{(1)}$. Based on the previous subsection, the following form is intuited:

$$a_1^{(1)} = -\frac{1}{2}r(R)\,\xi + q_1 r(R)^2 \left(u_1 \sin\Theta - u_2 \cos\Theta\right) + h_1(R)$$
$$a_2^{(1)} = q_2 r(R) \left(u_1 \cos\Theta + u_2 \sin\Theta\right) + h_2(R)$$
$$a_3^{(1)} = \lambda\xi \tag{12}$$
$$p^{(1)} = h_3(R)$$

As before, the Euler-Lagrange equations and the incompressibility condition give the two constants $q_1 = \frac{1}{8}\left(-2 + \lambda^3\right)$, $q_2 = \frac{1}{8}\left(2 + 3\lambda^3\right)$ and $h_1$, $h_2$, $h_3$ are a function of $R$. With the solutions for active strain $\mathbf{a}^{(0)}$ and $\mathbf{a}^{(1)}$, full expressions of them are given in the Appendix 4. The second order energy takes the form:

$$\mathcal{E} = \varepsilon^4 \int_0^L \mathrm{d}Z \int_S V_2\left(\boldsymbol{a}^{(0)}, \boldsymbol{a}^{(1)}; u_1, u_2, u_3, \xi\right) R\mathrm{d}R\,\mathrm{d}\Theta + \mathcal{O}\left(\varepsilon^5\right), \tag{13}$$

and we can transform it into this form:

$$V_2 R = A_1 \xi + A_2 \xi^2 + B_1 u_1 + B_2 u_1^2 + C_1 u_2 + C_2 u_2^2 + D_1 u_3 + D_2 u_3^2, \tag{14}$$

where $A_i$, $B_i$, $C_i$, and $D_i$ are functions of $R$ and $\Theta$.

## The method for determining deformation

Comparing with the energy of an extensible elastic rod (**Moulton et al., 2020a**; **Kirchhoff and Hensel, 1883**; **Mielke, 1988**; **Mora and Muller, 2003**; **Mora and Müller, 2004**), we recognize the classic extensional stiffness $K_0$, bending stiffness $K_1$ and $K_2$, torsional stiffness $K_3$ coefficients:

$$K_0 = \int_{S_K} A_2 dRd\Theta, \ K_1 = \int_{S_K} B_2 dRd\Theta, \ K_2 = \int_{S_K} C_2 dRd\Theta, \ K_3 = \int_{S_K} D_2 dRd\Theta, \tag{15}$$

where $A_2$, $B_2$, $C_2$, and $D_2$ are related to the shear modulus μ, so for a uniform material with no variations of shear modulus, $S_K$ represents the cross-section of the cylinder. But if not, we need to divide different regions to perform the integration.

We now focus on the intrinsic extension and curvature of the cylindrical object induced by the active strains, which requires the competition of the active forces with the stiffness of the cylindrical beam according to the relationships,

$$\widehat{\zeta} = 1 - H_0/K_0, \ \widehat{u}_1 = -H_1/K_1, \widehat{u}_2 = H_2/K_2, \ \widehat{u}_3 = -H_3/K_3, \tag{16}$$

each $H_i$ is calculated by:

$$H_0 = \frac{1}{2}\int_{S_H} A_1 dRd\Theta, \ H_1 = \frac{1}{2}\int_{S_H} B_1 dRd\Theta, \ H_2 = \frac{1}{2}\int_{S_H} C_1 dRd\Theta, \ H_3 = \frac{1}{2}\int_{S_H} D_1 dRd\Theta. \tag{17}$$

where $A_1$, $B_1$, $C_1$, and $D_1$ are related to the shear modulus μ, the fiber angles $\alpha$ and $\beta$, and the activation $g$. So, the integration region $S_H$ is divided into the different parts of the embryo which all contribute to the deformation.

To calculate the intrinsic Frenet curvature and torsion, we use the following equation:

$$\widehat{\kappa} = \sqrt{\widehat{u}_1^2 + \widehat{u}_2^2} \quad \text{and} \quad \widehat{\tau} = \frac{\widehat{u}_3}{\widehat{\zeta}}. \tag{18}$$

Finally, in the case of the activated left side muscles, we can calculate the intrinsic extension and curvatures using **Equation 16**, $\hat{u}_{1m} = 0$ and $\hat{u}_{3m} = 0$, so Frenet's intrinsic curvature and torsion **Equation 18**, $\hat{\kappa}_m = \hat{u}_{2m}$ and $\hat{\tau}_m = 0$. For the activated actin case, we can calculate the intrinsic extension and curvatures by **Equation 16**, $\hat{u}_{1a} = 0$ and $\hat{u}_{2a} = 0$, and the Frenet's intrinsic curvature and torsion **Equation 18**, $\hat{\kappa}_a = 0$ and $\hat{\tau}_a = \frac{\hat{u}_{3a}}{\hat{\zeta}_a}$. These quantities have been used to calculate the deformation images, as shown previously in **Figure 7**.

After obtaining all solutions, we can use these quantities to calculate the accumulated energy $W_c$ in the system after one contraction by *Equation 2*. By defining the activation ($g_m$ and $g_a$) and the energy conversion efficiency, we can obtain the activation ($g_{a1}$) of the actin bundles after a single contraction and further calculate the elongation.

All calculation results and parameters are presented in the Appendix.

## Acknowledgements

It is a pleasure to acknowledge fruitful discussions with Michel Labouesse and Kelly Molnar on experimental results. We also thank Alain Goriely for his help with the technical aspects of nonlinear elasticity. Both authors acknowledge the support of the contract EpiMorph (ANR-2018-CE13-0008). Anna Dai acknowledges the support of the CSC (China Scholarship Council), file No. 201906250173.

## Additional information

### Funding

| Funder | Grant reference number | Author |
|---|---|---|
| Agence Nationale de la Recherche | ANR-2018-CE13-0008 | Anna Dai<br>Martine Ben Amar |
| China Scholarship Council | 201906250173 | Anna Dai |

The funders had no role in study design, data collection and interpretation, or the decision to submit the work for publication.

### Author contributions

Anna Dai, Software, Formal analysis, Validation, Methodology, Writing – original draft, Writing – review and editing; Martine Ben Amar, Formal analysis, Supervision, Funding acquisition, Validation, Methodology, Writing – original draft, Writing – review and editing

### Author ORCIDs

Anna Dai ⓘ https://orcid.org/0000-0002-5655-8885
Martine Ben Amar ⓘ https://orcid.org/0000-0001-9132-2053

Reviewer #1 (Public Review): https://doi.org/10.7554/eLife.90505.3.sa1
Reviewer #2 (Public Review): https://doi.org/10.7554/eLife.90505.3.sa2
Author response https://doi.org/10.7554/eLife.90505.3.sa3

## Additional files

### Supplementary files

• MDAR checklist

### Data availability

All data generated or analysed during this study are included in the manuscript.

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

# Appendix 1

## Size and material parameters

**Appendix 1—table 1.** Parameters adopted in this work.

| | Normalized radii* | $R_1$=0.7 | $R_2$=0.8 | $R_2'$=0.96 | $R_3$=1 |
|---|---|---|---|---|---|
| **Geometry parameters** | **The location of muscles†** | $\theta_1 = \frac{2}{3}\pi$ $\theta_2 = \frac{5}{6}\pi$ | $\theta_3 = \frac{7}{6}\pi$ $\theta_4 = \frac{4}{3}\pi$ | $\theta_5 = \frac{5}{3}\pi$ $\theta_6 = \frac{11}{6}\pi$ | $\theta_7 = \frac{1}{6}\pi$ $\theta_8 = \frac{1}{3}\pi$ |
| | Actin part | $\mu_a = 5$ | | | |
| | Epidermis part | $\mu_e = 1$ | | | |
| **Material parameters (Shear modulus) ‡** | Muscles | $\mu_m = 100$ | | | |
| | Soft inner part | $\mu_i = 1/200$ | | | |

*The radii are extracted or inferred from the **Ben Amar et al., 2018**; **Lardennois et al., 2019**.

†The position and size of the muscles are inferred from the **Moerman and Williams, 2006**; **Lardennois et al., 2019**.

‡The units of the shear modulus are the **KPa** and is scaled by the epidermis one. It gives for the muscle, a value consistent with the muscle shear modulus proposed in **Denham et al., 2018**.

## Appendix 2

### Analytical model of the early elongation

The finite deformation gradient: $\mathbf{F_0} = Diag(r'(R), r(R)/R, \lambda)$, and $\mathbf{G_0} = Diag(1, g_0, 1)$, $0 < g_0 < 1$ in the actin layer, and $g_0 = 1$ in the part without actin bundles. To determine the finite strain $\mathbf{G_0}$, we converted the four muscle parts into thin layers attached to the epidermis in equal proportions to ensure the continuity of the model, and divided our model into four parts, see *Figure 1*. The actin layer($R_3 < R < R_2'$), epidermis layer($R_2 < R < R_2'$), and muscle layer($R_1' < R < R_2$) are considered as incompressible materials, and the inner part($0 < R < R_1'$) is compressible. The Neo-Hookean energy function is used for the incompressible parts:

$$W = \mu \left[ \frac{1}{2} \left( Tr \left( \mathbf{F_e} \mathbf{F_e}^T \right) - 3 \right) - q \left( \det \mathbf{F_e} - 1 \right) \right],$$ (19)

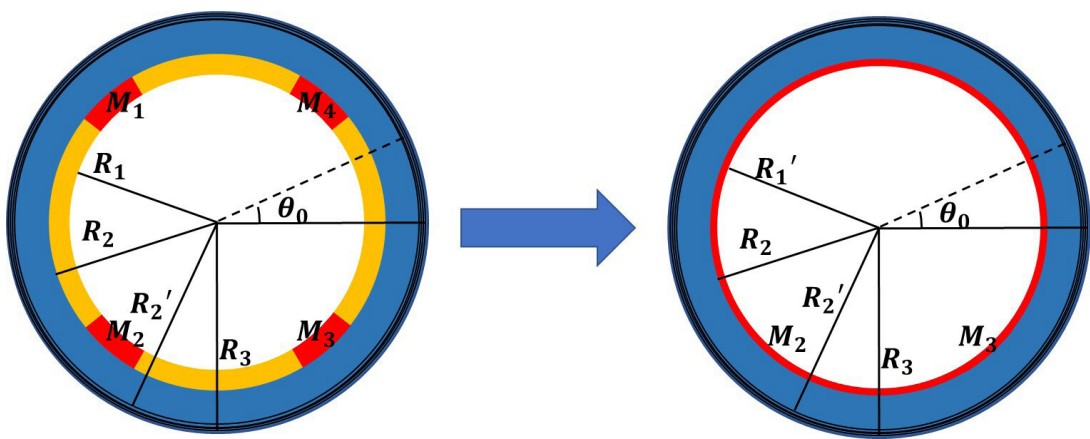

**Appendix 2—figure 1.** Simplified cross-sectional model with four scattered muscle sections simplified to thin layers ($R_1 = 0.7$, $R_1' = 0.768$, $R_2 = 0.7$, $R_2' = 0.96$).

According to the Euler-Lagrange equations, we can obtain the radius of the actin layer($r_a$), the epidermis layer($r_e$), and the muscle layer($r_m$):

$$r_a = \sqrt{\frac{g_0 R^2 + A}{\lambda}}, \quad r_e = \sqrt{\frac{R^2 + E}{\lambda}}, \quad r_m = \sqrt{\frac{R^2 + M}{\lambda}}.$$ (20)

where A, E. and M are constants, and the Lagrange multiplier $q$ in the actin, epidermis, and muscle layers:

$$q_a = \frac{-\log(R) + \frac{1}{2} g_0^2 \left( -\frac{A}{A + g_0 R^2} + \log\left(A + g_0 R^2\right) \right)}{g_0 \lambda} + C_a,$$ (21)

$$q_e = -\frac{\frac{E}{E + R^2} + 2\log(R) - \log\left(E + R^2\right)}{2\lambda} + C_e,$$ (22)

$$q_m = -\frac{\frac{M}{M + R^2} + 2\log(R) - \log\left(M + R^2\right)}{2\lambda} + C_m.$$ (23)

For the compressible part, we take the energy function form (*Holzapfel, 2000*):

$$W = \frac{\mu}{2} \left( Tr \left( \mathbf{F_e} \mathbf{F_e}^T \right) - 3 + \kappa \left( \det \mathbf{F_e} - 1 \right)^2 \right)$$ (24)

where $\kappa$ is a material constant. The radius of the inner part is:

$$r_i = \frac{aR}{\sqrt{\lambda}} \tag{25}$$

where $a$ is a constant.

By considering the boundary condition $\sigma_{rr} = 0$ on the outer border $R = 1$:

$$\sigma_{arr}(R = 1) = \mu_a \left[ \left( \frac{\partial r_a(1)}{\partial R} \right)^2 - q_a(1) \right] = 0 \tag{26}$$

the continuity of the radius:

$$r_a\left(R_2'\right) = r_e\left(R_2'\right), \quad r_e(R_2) = r_m(R_2), \quad r_m\left(R_1'\right) = r_i\left(R_1'\right), \tag{27}$$

and the continuity of the radial stresses $\sigma_{rr}$ in $R_2'$ and $R_2$:

$$\mu_a \left[ \left( \frac{\partial r_a(R_2')}{\partial R} \right)^2 - q_a\left(R_2'\right) \right] = \mu_e \left[ \left( \frac{\partial r_e(R_2')}{\partial R} \right)^2 - q_e\left(R_2'\right) \right], \tag{28}$$

$$\mu_e \left[ \left( \frac{\partial r_e(R_2)}{\partial R} \right)^2 - q_e(R_2) \right] = \mu_m \left[ \left( \frac{\partial r_m(R_2)}{\partial R} \right)^2 - q_m(R_2) \right]. \tag{29}$$

Using the above equations, we can obtain expressions for the constants $E$, $M$, $C_a$, $C_e$, and $C_m$.

We replace all size and material parameters and $\kappa = 100$, $\lambda = 1.8$ in the last condition $\sigma_{mrr}\left(R_1'\right) = \sigma_{irr}\left(R_1'\right)$:

$$\mu_m \left[ \left( \frac{\partial r_m(R_1')}{\partial R} \right)^2 - q_m\left(R_1'\right) \right] = \mu_i \sigma_{irr}\left(R_1'\right), \tag{30}$$

where $\sigma_{irr}$:

$$\sigma_{irr} = \frac{2}{\det \mathbf{F_e}} \left( (\det \mathbf{F_e})^2 \frac{\partial W}{\partial (\det \mathbf{F_e})^2} + \frac{\partial W}{\partial \left( tr(\mathbf{F_e}\mathbf{F_e}^T) \right)} r_i'(R)^2 \right). \tag{31}$$

and obtain the relation with constant $A$ and $g_0$.

Finally, by prescribing a zero traction condition on the top of the cylinder, the muscle part was noticeably considered inextensible, so there was no stress on the top:

$$\int_{R_2'}^1 \sigma_{azz} r_a r_a' dR + \int_{R_2}^{R_2'} \sigma_{ezz} r_e r_e' dR + \int_0^{R_1'} \sigma_{izz} r_i r_i' dR, \tag{32}$$

where:

$$\sigma_{azz} = \lambda^2 - q_a, \quad \sigma_{ezz} = \lambda^2 - q_e, \tag{33}$$

$$\sigma_{izz} = \frac{2}{\det \mathbf{F_e}} \left( (\det \mathbf{F_e})^2 \frac{\partial W}{\partial (\det \mathbf{F_e})^2} + \frac{\partial W}{\partial \left( tr(\mathbf{F_e}\mathbf{F_e}^T) \right)} \lambda^2 \right), \tag{34}$$

all solutions $g_0 = 0.88$, $A = 0.08$, and $r_a(1) = 0.73$ can be determined. The result is also in good agreement with the experimental data (**Vuong-Brender et al., 2017a**; **Ben Amar et al., 2018**).

## Appendix 3

### Time-scale for myosin detachment

The time required for non-muscle myosin detachment is estimated to be $\tau_0 = 10s$ for free actomyosin filaments. If the actin filament is subjected to an external load perpendicular to its axis, detachment can be facilitated, or on the contrary, inhibited, see *Howard, and Clark,, 2002*, page 169–170. In the present case, the stresses acting in the radial direction of an actin bundle are compressive and thus will retard the detachment. This energy must be compared to the energy of detachment of all myosin motors from the bundle. The corresponding elastic energy associated with the radial deformation for an actin cable of length $l_a$ with radius $r_b = 0.05\mu m$ estimated from *Lardennois et al., 2019* and shear modulus $\mu_a = 5\ KPa$ is given by $1/2\mu_a(1 - g_0(t))\pi l_a r_b^2 = 210^{-11} l_a\ J$. This result must be compared to the individual energy of detachment times the number of myosin motors on a cable. This number is difficult to determine but an estimate is given by the length of the cable $l_a$ divided by the distance between 2 myosin anchoring sites, which is about $5nm$ while the attachment energy per motor is about $6k_bT$. These values are for skeletal muscle myosins (*Howard, and Clark,, 2002*) and must be taken with caution. Nevertheless, the order of magnitude of the detachment energy for a collection of myosin heads from actin cable can be estimated to be $4.8 l_a 10^{-12}J$. Then the ratio between the two quantities is of the order $4(1 - g_0(t))$, which explains that the time scale of debonding for a cable under compressive stress in the direction orthogonal of its axis is then:

$$\tau_{deb} = \tau_0 e^{p_3(1-g_0(t))} \tag{35}$$

where $p_3$ is a positive constant of order one that cannot be predicted exactly. This time scale justifies the exponential correction in *Equation 3* of the manuscript.

## Appendix 4

### Modeling in the absence of pre-strain

The paper discusses the case with pre-strain. Here, are more details about the case without pre-strain ($\mathbf{G_0} = \mathbf{I}$). Up to the lowest order, the solution of the Euler–Lagrange equations is obviously given by $\mathbf{a}^{(0)} = (R, \Theta, 0)$, and $p^{(0)} = 1$, so there is no deformation and the zero and first order energies vanish. At order $\mathcal{O}\left(\varepsilon^2\right)$ of the elastic energy, the solutions for the active strain components are

$$
\begin{aligned}
a_1^{(1)} &= -\frac{R}{2}\xi - \frac{R^2}{8}\left(u_1 \sin\Theta - u_2 \cos\Theta\right) + f_1\left(R\right), \\
a_2^{(1)} &= \frac{5R}{8}\left(u_1 \cos\Theta + u_2 \sin\Theta\right) + f_2\left(R\right), \\
a_3^{(1)} &= 0, \\
p^{(1)} &= f_3\left(R\right).
\end{aligned}
\tag{36}
$$

where $f_{1,2}$ are functions related to the active stress $g$ and the fiber angles $\alpha$ and $\beta$:

$$
\begin{aligned}
f_1\left(R\right) &= \frac{gR}{2}, \\
f_2\left(R\right) &= g\log\left(R\right)\sin^2\alpha \sin\left(2\beta\right), \\
f_3\left(R\right) &= 2g\mu\log\left(R\right)\sin^2\alpha \sin\left(2\beta\right).
\end{aligned}
\tag{37}
$$

The stiffness coefficients:

$$
\begin{aligned}
A_2 &= \frac{9\mu R}{8}, \\
B_2 &= \frac{81}{128}\mu R^3 \sin^2\Theta, \\
C_2 &= \frac{81}{128}\mu R^3 \cos^2\Theta, \\
D_2 &= \frac{\mu R^3}{2}.
\end{aligned}
\tag{38}
$$

The deformation coefficients:

$$
\begin{aligned}
A_1 &= -\frac{1}{8}g\mu R\left[4 + 5\cos\left(2\alpha\right) + 2\cos\left(2\beta\right)\left(-1 + 2\log\left(R\right)\right)\sin^2\alpha\right], \\
B_1 &= -\frac{1}{16}g\mu R^2\left[10 + 17\cos\left(2\alpha\right) + 2\cos\left(2\beta\right)\left(-1 + 14\log\left(R\right)\right)\sin^2\alpha\right]\sin\Theta, \\
C_1 &= \frac{1}{16}g\mu R^2\left[10 + 17\cos\left(2\alpha\right) + 2\cos\left(2\beta\right)\left(-1 + 14\log\left(R\right)\right)\sin^2\alpha\right]\cos\Theta, \\
D_1 &= -g\mu R^2 \cos\beta \sin\left(2\alpha\right).
\end{aligned}
\tag{39}
$$

### Modeling with pre-strain

The first order solutions of the theory with the case of pre-strain, $h_{1,2,3}\left(R\right)$, are related to the activation $g$ and the fiber angles $\alpha$ and $\beta$:

$$
\begin{aligned}
h_1\left(R\right) &= \frac{R\left(c + g\right)}{2\sqrt{\lambda}}, \\
h_2\left(R\right) &= g\log\left(R\right)\sin^2\alpha \sin\left(2\beta\right), \\
h_3\left(R\right) &= \frac{2\mu\log\left(R\right)\left[c + g\cos\left(2\beta\right)\sin^2\alpha\right]}{\lambda}.
\end{aligned}
\tag{40}
$$

The stiffness coefficients are:

$$
\begin{aligned}
A_2 &= \left(0.278 + 1.689\mu\right)R, \\
B_2 &= 1.294R^3\left[-0.405 + 2.389\mu + \left(-0.176 + \mu\right)\cos\left(2\Theta\right)\right], \\
C_2 &= -1.294R^3\left[0.405 - 2.389\mu + \left(-0.176 + \mu\right)\cos\left(2\Theta\right)\right], \\
D_2 &= 0.9\mu R^3.
\end{aligned}
\tag{41}
$$

The deformation coefficients are:

$$A_1 = \mu R \left[ 0.417c - 0.139g - 6.48g\cos^2\alpha - 0.556c \log(R) \right.$$
$$\left. + g \left( 0.556\cos^2\beta - 0.556 \cos(2\beta) \log(R) \right) \sin^2\alpha \right].$$

$$B_1 = R^2 \left\{ g \left( -0.828 + 4.830\mu \right) \cos\beta \log(R) \sin^2\alpha \sin\beta \cos\Theta + 0.198c \sin\Theta \right.$$
$$+ \left[ 0.198g + 0.910c\mu + 1.307g\mu - 4.830g\mu\cos^2\alpha - 1.225c\mu \log(R) \right.$$
$$\left. + g\mu \left( -0.397\cos^2\beta - 1.225 \cos(2\beta) \log(R) \right) \sin^2\alpha \right] \sin\Theta \Big\}.$$

$$C_1 = R^2 \left\{ g \left( -0.828 + 4.830\mu \right) \cos\beta \log(R) \sin^2\alpha \sin\beta \sin\Theta - 0.198c \cos\Theta \right.$$
$$+ \left[ -0.198g - 0.910c\mu - 1.307g\mu + 4.830g\mu\cos^2\alpha + 1.225c\mu \log(R) \right.$$
$$\left. + g\mu \left( 0.397\cos^2\beta + 1.225 \cos(2\beta) \log(R) \right) \sin^2\alpha \right] \cos\Theta \Big\}.$$

$$D_1 = -g\mu R^2 \cos\beta \sin(2\alpha).$$

$$(42)$$

## Appendix 5

### Energy transformation calculations

The energy transformation process is depicted in *Figure 1*.

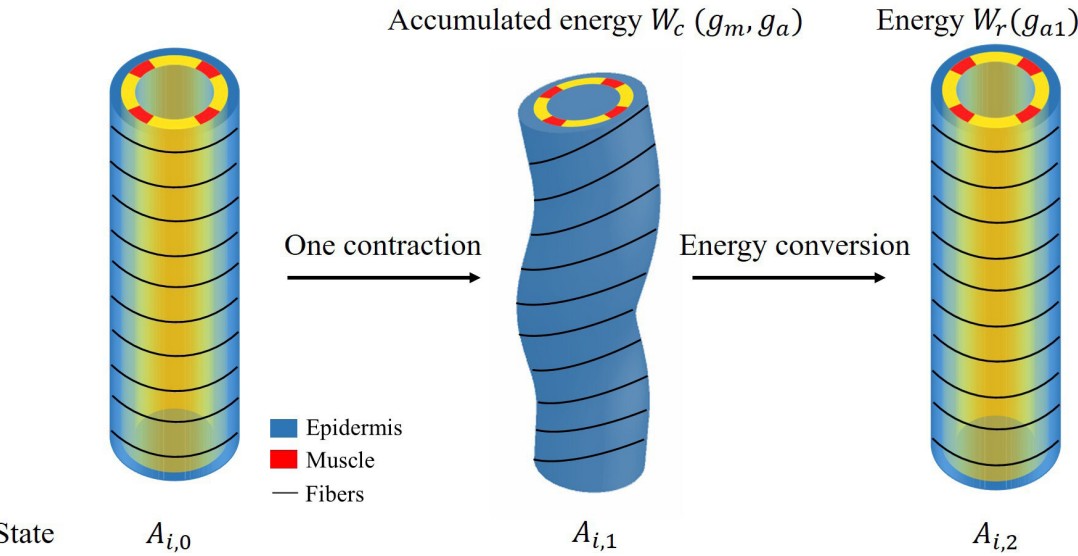

Accumulated energy $W_c\,(g_m, g_a)$ Energy $W_r(g_{a1})$

One contraction Energy conversion

■ Epidermis
■ Muscle
— Fibers

State $A_{i,0}$ $A_{i,1}$ $A_{i,2}$

**Appendix 5—figure 1.** Schematic diagram of energy conversion.

To obtain the elongation $\delta\zeta_i$ after each muscle contraction, we need to calculate the energy and the total energy has the following form:

$$\mathcal{E} = \varepsilon^2 \int_0^L dZ \int_S \left( V_0 + \varepsilon V_1 + \varepsilon^2 V_2 \right) R\, dR\, d\Theta + O\left( \varepsilon^5 \right) \tag{43}$$

where the integration domain $S$ refers to each part of the cylinder with a different shear modulus $\mu$, so the model must be divided into three different parts for integration.

The final part is the energy conversion per unit volume:

$$\mathcal{E}_I = \int_S \left( \varepsilon V_1 + \varepsilon^2 V_2 \right) R\, dR\, d\Theta \tag{44}$$

where $V_1$ is not the total first order energy, we consider only the energy induced by the activation of actomyosin $g_a$ and muscle $g_m$. After obtaining solutions $\mathbf{a}^{(0)}$, $\mathbf{a}^{(1)}$ and deformations from *Equations 11 and 12*, the accumulated energy during the contraction period $W_c$, which we define, is:

$$
\begin{aligned}
W_c &= \int_S \left( \varepsilon V_1 + \varepsilon^2 V_2 \right) R\, dR\, d\Theta \\
&= \varepsilon \left( -10.70 g_a - 7.73 g_m \right) + \varepsilon^2 \left[ \left( -3.31 + 26.75 g_a \right) g_a + 11.95 g_m^2 \right].
\end{aligned}
\tag{45}
$$

When the muscles are relaxed and only actomyosin is activated, the total increase in volumetric energy is $W_r$:

$$
\begin{aligned}
W_r &= \int_S \left( \varepsilon V_1 + \varepsilon^2 V_2 \right) R\, dR\, d\Theta \\
&= -0.41 \varepsilon g_{a1} + \varepsilon^2 \left( -5.19 + 6.96 g_{a1} \right) g_{a1}.
\end{aligned}
\tag{46}
$$

By calculating the energy conversion, we obtain $g_{a1} = -0.66$ at the beginning of the late elongation phase, *Figure 7* of the manuscript shows the elongation for each contraction and total elongation varies with time.

