## [Editor Report · eLife assessment]

Using continuum theory of elastic solids the authors present evidence that periodic muscle contraction leads to elongation of *C. elegans* embryos by storing elastic energy that is subsequently released by extending the embryo's long axis. This **important** finding could apply to other developmental processes and be exploited in soft robotics. The presented evidence is **convincing** on the phenomenological level adopted in the work. How bending energy is converted into elongation on a more microscopic level remains to be worked out.

---

## [Referee Report · Reviewer #1 (Public Review)]

The authors have made a novel and important effort to distinguish and include different sources of active deformations for fitting C elegans embryo development: cyclic muscle contractions and actomyosion circumferential stresses. The combination and synchronisation of both contributions are, according to the model, responsible for different elongation rates, and can induce bending and torsion deformations, which are a priori not expected from purely contractile forces. The model can be applied to other growth processes in initially cylindrical shapes.

The tilt of the fibers is an important assumption of the model. However, fiber direction in Figure 3B is not fully clear for explaining the tilting. The fiber in 3B has not very much in common with the fibers in the color part of the figure. Also, is vector m supposed to be tangent to the fiber? In the figure does not seem to be so. It should be expected that alpha is a consequence of the deformation, not as an input parameter, as it seems in the tests of Figure 6A. How is the value of alpha chosen? According to Figure 6, torsion is expected for alpha>0, but for beta=pi/2 and alpha>0 no torsion may be obtained. In fact, it seems that torsion should appear when cos(beta)*sin(alpha)>0. As a consequence, value of beta should be given in Figure 6. Can the amount of torsion be tested as a function of alpha and beta?

The transfer of energy and deformation is a very interesting aspect of the paper, and also crucial for the model and predicting elongation. However, the modelling of this transfer remains very obscure and only explained in the Appendix. Some more details on how the transfer is selected should be given in the main text. Can the transfer of energy interpreted as a change of the relaxed reference configuration? Once a ratio of the energy transferred is fixed, the assumption on elongation distribution should be stated. (Uniformly? ) The authors should also define in the main text the factor g_a1, and explain how this value is computed from condition W_c=W_r .

Given the convoluted shape of the embryo in the egg, contact may be a crucial mechanism for determining growth and torsion. The model does not include this contact, and this limitation should be reflected in the article.

Minor comment:

-Line 300: "we determine the optimal values for the activation parameters". the optimal with respect to which objective? Norm of difference between experimental and computational displacements? How this is quantified needs to be specified.

---

## [Referee Report · Reviewer #2 (Public Review)]

Summary:

During *C. elegans* development, embryos undergo elongation of their body axis in absence of cell proliferation or growth. This process relies in an essential way on periodic contractions of two pairs muscles that extend along the embryo's main axis. How contraction can lead to extension along the same direction is unknown.

To address this question, the authors use a continuum description of a multicomponent elastic solid. The various components are the interior of the animal, the muscles, and the epidermis. The different components form separate compartments and are described as hyperelastic solids with different shear moduli. For simplicity, a cylindrical geometry is adopted. The authors consider first the early elongation phase, which is driven by contraction of the epidermis, and then late elongation, where contraction of the muscles injects elastic energy into the system, which is then transferred into elongation. The authors get elongation that can be successfully fitted to the elongation dynamics of wild type worms and two mutant strains.

Strengths:

The work proposes a physical mechanism underlying a puzzling biological phenomenon. The framework developed by the authors could be used to explain phenomena in other organisms and could be exploited in the design of soft robots.

Weaknesses:

(1) The manuscript is hard to read without being very familiar with continuum descriptions of elastic media. This might make the work difficult to access for biologists. This is a real pity because the findings are potentially of great interest to developmental biologists and engineers alike.

(2) The discussion of the worm's mechanical properties could go deeper. The authors hardly justify their assumptions.

---

## [Author Response]

The following is the authors’ response to the original reviews.

**Recommendations for the authors:**

**Reviewer #1 (Recommendations For The Authors):**
Line 144, after eq. (1). Vectors d_i need to be defined. Are these the mapping of vectors e_i due to the active deformation? It would be useful to state then that d_3 is aligned with r'.

Thank you for your suggestion, and the definition has been added to lines 146-149 for a better understanding of the model.

Line 144.Authors state a_i(0,0,Z)=0. Shouldn't this be true also for any angle, i.e., a_i(0,Theta, Z)=0?

Thank you, we have revised it in line 144.

Line 156. G_0 is defined as Diag(1,g_0(t), 1), which seems to be using cylindrical coordinates. Previously, in line 147, vector argument X of \chi is defined with Cartesian coordinates (X,Y,Z). Shouldn't these be also cylindrical?

We are very sorry for this error, our initial configuration is defined with cylindrical coordinates, we have revised it in the manuscript line 151.

Line 162. "where alpha and beta lie in the range [-pi/2, pi/2]" has already been indicated.

Thank you for your mention, we have deleted duplicate information in line 166.

Line 171. W is defined as the strain energy density, while in equation (2), symbol W is the total energy (which depends on the previous W). Letters for total elastic and strain energy must be distinguished.

Thank you, we have changed the letter for total energy in Eq.(2).

Line 176. "we take advantage of the weakness of" -> "we take advantage of the small value of".

We have revised it in line 179.

Line 177. Why is there a subscript i in p_i? If these do not correspond to penalty p, but to parameters in eqn (3), the latter should have been introduced before this line.

We have revised this error in line 180.

Line 186. "as the overall elongation \zeta". This parameter, axial extension, has not been defined yet.

Thank you for your mention, the definition of \zeta is now given in line 146.

Figure 4. Why are the values of g_0 from the elastic model and equations (30)-(32) so non-smooth? Clarify what is being fit and what is the input in the latter equations. Final external radius R_3? Final internal radius R_1'?

(1) To mimic the embryo, we consider a multi-layered cylindrical body so that the shear modulus of each layer is different. The continuity of both deformations and stresses is imposed (see Eq.(26)-Eq.(30)). This is the usual treatment for complex morpho-elastic systems. Obviously, $g_0$ originates from the actomyosin cortex so it appears only in the corresponding layer. Finally, all physical quantities such as deformations and stresses must be continuous.

(2) The final outer radius is R_3, which represents the outer radius of *C. elegans* embryos. In addition to R_3, what we need to consider in this model are R_1’=0.7, R_1’=0.768, R_2=0.8 and R_2’=0.96, these definitions have been added in the caption of Appendix 2—figure 1.

Line 663, equation (19). Parameter mu is multiplying penalisation term with p, while in equation (2) mu is only affecting the elastic part.

These two different ways of expressing the energy function will ultimately affect the value of p, but the two p are not the same quantities, so they will not affect our results. To avoid misunderstandings, we will replace p in equation (19) with q.

**Reviewer #2 (Recommendations For The Authors):**
As mentioned in my public summary, I find the writing really not adequate. I provide here a list of specific points that the authors should in my opinion address. As a general comment, I would delete many instances of 'the'.First, here are figures and whole paragraphs that do not seem to bring anything to the understanding of the phenomenon of *C. elegans* elongation, notably, Figs. 2, 3C-H, 5m, and 6. Figures 6G and 7 are the only figures containing results it seems. Some elements of the figures are repeated, for example, the illustration of the system's cross-section in Figs 3 and 5.

Thank you for your suggestion, we have made some adjustments to our images to remove some of the duplicate information.

Second, and this is my most important criticism: the mechanism of elongation by releasing elastic stress introduced by muscle contraction is not explained in clear terms anywhere in the text. At least, I was unable to understand it. On p 10 you write "This energy exchange causes the torsion-bending energy to convert into elongation energy, (...)" How this is done is not explained. I assume that the reference state is somehow changed through muscle contraction. The new reference state probably has a longer axis than the one before, but this would then be a plastic deformation and not purely elastic as claimed by the authors (ll 76: "This work aims to answer this paradox within the framework of finite elasticity without invoking cell plasticity (...)"). Is torsion important for this process or is it 'just' another way to store elastic energy in the system?

We perfectly explain most of the exchange of energy between bending, torsion and elongation: indeed, we quantify all aspects of this transformation as the elastic elongation energy, and the dissipation processes which will cost energy. The dissipation evaluated here concerns the rotation of the worm due to the muscle geometry and the viscous friction at the inner surface of the egg. Torsion seems to appear in the late stages and only in some cases. As we show, it comes from a torque induced by the muscles which are not vertical. vertical. Finally, our quantitative predictions of the modelling which recovers most of the experimental published results.

Third, there are a number of strange phrasings and the notation is not helpful in places.

We feel sorry for that, the manuscript is now more precise.

Fourth, the title promises to explain how cyclic muscle contractions reinforce acto-myosin motors. I can't see this done in this work.

The fact that the acto-myosin is reorganized between two sequences of contraction justifies the title. The complete reorganization of the actomyosin network would require a chemico-mechanical model that is not achieved here, perhaps in future work as data become available.

In addition:

We have chosen to respond globally rather than point by point to the referee’s recommendations.

Typographic errors and vocabulary

All English corrections and typos are now included in the main text.

Figures and captions:

Figures and captions have been improved.

Figure 1: Make the caption and the illustration more coherent. For example, only two cell types are distinguished; in the caption, you mention lateral cells, in the sketch seam cells. What is the difference between acto-myosin and muscle contraction? Muscle contraction is also auto-myosin-based.

(1) The caption for Fig.1 is revised.

(2) From a mechanical point of view, actomyosin bundles in C elegans are orthoradial, whereas muscles are essentially parallel to the main axis of the body are essentially parallel to the main axis of the body, so the geometry is completely different and of extreme importance for deformation. Muscle contractions are quasi-periodic, we do not know the dynamics of the attached molecular motor of myosin. So of course, both contain actin and myosin (not exactly the same proteins), but our model is sensitive to more macroscopic properties.

Figure 2: I do not find this figure helpful. I might expect such a figure in a grant proposal, but much less in an article.

Figure 2 shows the strategy of our work, we hope that readers can see at a glance what kind of analysis has been done through this figure: since our work is divided into several parts, readers can also unravel the logic through this scheme after reading the whole manuscript. So, this diagram is a guide, and it may be helpful and necessary.

Figure 3: Figure 3 A, right: What is the dashed line? B You indicate fibers, but your model does not contain fibers, does it? How do I get from the cube to the deformed object? What is the relation of C-H with the rest of the work? Furthermore, you mention seam cells in Fig. 1, but they are absent here. Why can you neglect them? Why introduce them in the first place? E What is a plant vine? F-H What rods are you referring to? Plants do not have muscles, right?

We have modified this figure, and the original Figure 3 now corresponds to Figures 3 and 4.

(1) The dashed line is the centerline after deformation.

(2) The referee is wrong: our model represents the fibers by a higher shear modulus for the actomyosin cortex and for the muscles (see Table Appendix 1) and G_1 reflects the activities of the muscle and actin fibers.

(3) The cube in Figure 3 is a mathematical 3D volume element that is subjected to stresses. Hyperelasticity modelling is based on such a representation.

(4) C-H(new version: Fig.4 A-F): These images show similar deformations: bending and torsion as our *C. elegans* study. These figures indicate that such deformations are quite common in nature, even if the underlying mechanism is different.

(5) This is a point we have already mentioned: we ignore the difference between the different types of epidermal cells and average their role in the early and second stages of elongation.

(6) The plant vine is the 'botanical vine', see Goriely's article and book.

(7) F-H(new version: Fig.4 D-F) do not have fixed rods, we set a curvature and torsion to fit the actual biological behavior.

(8) Plants do not have muscles, but they grow, and our formalism for growth, pre-strain and material plasticity is very similar to the hyper-elasticity formalism.

Figure 4: Fig .4 A: "The central or inner part (0 < 𝑅 < 𝑅2, shear modulus 𝜇𝑖) except the muscles which are stiffer." I do not understand.

In the new version, this figure corresponds to Fig.5. The shear modulus of the intrinsic part is very small, but the muscles are harder so we have to consider them separately, we have revised this sentence to avoid misunderstanding.

Figure 5: Fig 5 A and D: The schematic of the cross-section has appeared already in the previous figure. No need to repeat it here. The same holds for the schematic of the cylindrical embryo. Caption: "But, the yellow region is not an actual tissue layer and it is simply to define the position of muscles." Why do you introduce the yellow region at all? I do not think that it clarifies anything. "Deformation diagram, when left side muscles M_1 and M_2." Something seems to be missing here. Similarly in the next sentence. "the actin fiber orientation changes from the 'loop' to the 'slope'" Do the rings break up and form a helix?

In the new version, this figure corresponds to Fig.6.

(1) We have made revisions to these figures.

(2) The yellow part can show the accurate location of four muscles, which is important for our model and further calculations.

(3) We have revised this sentence in the caption of Fig. 6.

(4) Actin rings do not change to a helix pattern, they will be only sloping.

Figure 6: Fig 6 A-C These panels do not go beyond Fig 5B. Fig 6D: what are these images supposed to show? They are not really graphs, but microscopy images. The caption is not helpful to understand, what the reader is supposed to see here. Fig 6F: do you really want to plot a linear curve?

In the new version, Fig.5 and Fig.6 respectively correspond to Fig.6 and Fig.7.

(1) Fig.6 shows the simulated images, and Fig.7 A-C is the real calculation results, they are different.

(2) Fig.7 D can show the real condition during *C. elegans* late elongation, here, we would like to show the torsion of the *C. elegans*.

(3) Yes, it is our result.

Discussions concerning the biological referee questions:Ll 75: “how the muscle contractions couple to the acto-myosin activity" Again I find this misleading because muscle contraction relies on auto-myosin activity. Probably, you can find a better expression to refer to the activity of the actomyosin network in the epidermis. Do you propose any mechanism for how muscle contraction increases epidermal contractility? This does not seem to be the mechanism that you propose for elongation, is it?

The actomyosin activity will not stop because of the muscle contraction. Obviously, these two processes cannot be independent. The energy released by a muscle contraction event can and must contribute to the reorganization of the actomyosin network that occurs during the elongation process. Indeed, despite the fact that the embryo elongates, the density of actin cables appears to be maintained, which automatically requires a redistribution of actin monomers. We propose a scenario in which muscle contraction increases actomyosin contractility via energy conversion. We show that after unilateral contraction there is an energy release for this once all dissipation factors are eliminated. We invite the reviewer to re-examine Figure 2 and invite biologists to seriously evaluate the density of molecular motors attached to the circumferential actin cable throughout the stretch process.

Ll 133: "we decide to simplify the geometrical aspect because of the mechanical complexity" This is hardly a justification. Why is it appropriate?

Yes, we would like to offer the reader the simplest modelling with a limiting technicity and a limited number of unknown parameters.

L 135: "active strains" Why not active stress?

The two are equivalent, the choice is dictated by the simplicity of deriving quantitative results for comparison with experiments.

L 170: "hyperelastic" Please, explain this term.

It is the elasticity of very soft samples subjected to large deformations. For classic references, see the books of Ogden, Holzapfel and Goriely, all of which are mentioned in our paper.

Major criticismEq. 3 and Ll 227: "𝑝1 is the ratio between the free available myosin population and the attached ones divided by the time of recruitment" Why is the time of recruitment the same for all motors? "inverse of the debonding time" Is it the same as the unbinding rate? Why use the symbol p_2 for it? What is p_3?

The model proposed to justify the increase in the activity of the actomyosin motors during the first phase is a mean-field model: thus all quantities are averaged: we are not considering the theory of a single molecular motor, but a collection in a dynamic environment, so we do not need stochasticity here. Equation (3) concerns the compressive pre-strain, which by definition is a quantity varying between $0$ and $1$ and $X_g=1-G$. ... The debonding time is not the same as the debonding rate. The term $p_3$ indicates saturation and is derived from the law of mass action. The good agreement with the experimental data is shown in Fig.5 (A) and (B). An equivalent model has been developed by (M. Serra et al.).

Serra M, Serrano Nájera G, Chuai M, et al. A mechanochemical model recapitulates distinct vertebrate gastrulation modes[J]. Science Advances, 2023, 9(49)

Ll 275: "This energy exchange causes the torsion-bending energy to convert into elongation energy, leading to a length increase during the relaxation phase, as shown in Fig.1 of Appendix 5." You have posed the puzzle of how contraction leads to elongation, and now that you resolve the puzzle, you simply say that torsion and bending energy are converted into elongation. How? Usually, if I deform an elastic object, it will return to its original configuration after releasing the external forces. Why is this not the case here?Furthermore, the central result of your work is presented in an Appendix!?

We agree with the referee that an elastic object will return to its initial configuration by releasing stress, i.e. by giving up its accumulated elastic energy to the environment. But the elastic energy has to go somewhere, such as heat. We do not dare to say that the temperature of the worm increases during the muscle contractions.

In fact, the referee's comment also assumes that full relaxation of the stresses is possible, so the object is not a multi-layered specimen and/or it is not enclosed in a box. Most living species are under stress, usually called residual stress. Our skin is under stress. Our fingerprints result from an elastic instability of the epidermis, occurring on foetal life as our brain circumvolutions or our vili. . So, it is obvious that stresses are maintained in multilayered living systems. Closer to the case of *C. elegans*, the existence of stresses has been demonstrated by experiments with laser ablation fractures in the first stage. The fact that the fractures open proves the existence of stress: if not, there is no opening and only a straight line.

Ll 379: "Although a special focus is made on late elongation, its quantitative treatment cannot avoid the influence of the first stage of elongation due to the acto-myosin network, which is responsible for a prestrain of the embryo." This statement is made repeatedly through the manuscript, but I do not understand, why you could not use an initial state without pre-strain.

This is the basic concept of hyperelasticity. The reference state must be free of stress, so we cannot evaluate the first muscle contraction without treating the first elongation stage.

Grammar, vocabulary and writing errorsll 31: "the influence of mechanical stresses (...) becomes more complex to be identified and quantified" Is the influence of mechanical stress too complex or too difficult to be identified/quantified?

We have revised it in line 31, “The superposition of mechanical stresses, cellular processes (e.g., division, migration), and tissue organization is often too complex to identify and quantify.”

Ll 41: "The embryonic elongation of *C. elegans* represents an attractive model of matter reorganization without a mass increase before hatching." Maybe "Embryonic elongation of *C. elegans* before hatching represents an attractive model of matter reorganization in the absence of growth.".

We have revised it in line 41.

L 42: "It happens after the ventral enclosure (...)" Maybe "It happens after ventral enclosure (...)".

We have revised it in line 42.

Ll 52: "The transition is well defined since the muscle participation makes the embryo rather motile impeding any physical experiments such as laser ablation (...)" Ablation of what?

We have revised it in line 53:The transition is well defined, because the muscle involvement makes the embryo rather motile, and any physical experiments such as laser fracture ablation of the epidermis, which could be performed and achieved in the first period (Vuong-Brender et al., 2017a), become difficult,.

Ll 59: "a hollow cylinder composed of four parts (seam and dorso-ventral cells)" It is not clear, what the four parts are - in the parenthesis, two are mentioned.

We have revised it in line 59. Fig.1 shows the whole structure, dorsal, ventral and seam cells form four parts of the epidermis.

L 78: "several important issues at this stage remain unsettled" At which stage?

It means the late elongation stage, we have added this information in line 78.

Ll 85: "but how it works at small scales remains a challenge." Maybe "but how it works at small scales remains to be understood.".

We have revised it in line 86.

Ll 99: "the osmolarity of the interstitial fluid" The comes out of the blue. Before you only talked about mechanics, why now osmolarity? Also, the interstitial fluid is only mentioned now. It is important for the dissipative effects that you discuss later, right? If yes, then you should probably introduce it earlier.

For a better understanding, we have change osmolarity into viscosity in line 99.

l 120: "The cortex is composed of three distinct cells" Maybe "distinct cell types".

Thank you, and we have revised it in line 120.

L 121: "cytoskeleton organization and actin network configurations" What is the difference between cytoskeleton organization and actin network configuration? Also, either both should be plural or both singular, I guess.

(1) Cytoskeleton (which involves microtubules) forms the epidermis of *C. elegans* embryos, and the actin network surrounds the epidermis.

(2) Thank you for your suggestion, we have revised it in line 121.

L 130: "which will be introduced hereafter" Maybe "which will be used hereafter".

We have revised it in line 130.

Ll 148: "The geometric deformation gradient" You usually denote vectors in bold face, so \chi should be bold, right? Define d_i in Eq.(1).

Yes, we have added this information in line 147.

L 172: "auxiliary energy density" Please, explain this term.

We have changed "auxiliary energy density" into "associated energy density" in line 175. Energy density is the amount of energy stored in a given system or region of space per unit volume, the associated energy density in our manuscript can help us to do some calculations.

Ll 188: "Similar active matter can be found in biological systems, from animals to plants as illustrated in Fig.3(C)-(E), they have a structure that generates internal stress/strain when growing or activity. (...)" Why such a general statement during the presentation of the results? The second part of the sentence seems to be incomplete.

Answers: We would like to show our method is general, and can be used in many situations. We have revised the wrong sentence in line 192.

Ll 243: "a bending deformation occurs on the left for active muscles localized on left" Maybe "bending to the left occurs if muscles on the left are activated".

Thank you, we have revised it in line 247.

L 250: "we assume them are perfectly synchronous" Maybe "we assume them to contract simultaneously".We have revised it in line 252.L 258: "the muscle and acto-myosin activities are assumed to work almost simultaneously." Before it was simultaneously, now only almost!? What does almost mean?

Sorry, we would like to express the same meaning in theses two sentences, we have deleted the word ‘almost’ in line 261.

Ll 294: "one can hypothesize several scenarios" After that, only one scenario is described it seems.

Thank you, we have revised this sentence in line 299.

L 341: "and then is more viscous than water" Maybe "and that is more viscous than water".

We have revised it in line 345.

L 373: "before the egg hatch" Maybe "before the embryo (or larva) hatches"?

We have revised the sentence in line 367.

L 409: "elephant trunk elongated" maybe "elephant trunk elongation".

We have revised it in line 412.

Ll 417: "As one imagines, it is far from triviality (...)" Does this remake help in any way to understand better *C. elegans* elongation? Also maybe "it is far from trivial".

We have revised it in line 423.

Ll 428: "can map the initial stress-free state B_0 to a state B_1, which reflects early elongation process" Maybe: "maps the initial stress-free state B_0 to a state B_1, which describes early elongation".

We have revised it in line 428.

L 429: "After in the residually stressed (...)" Maybe "Subsequently, we impose an incremental strain filed G_1 that maps the state B_1 to the state B_2, which represents late elongation".

We have revised it in line 429.

l 763: "Modelling details of without pre-strain case" Maybe "Case without pre-strain" or "Modelling in the absence of pre-strain" Similarly for l 784.

We have revised them in line 763 and line 784.

Some questions of definition and understandingLl 71: "We can imagine that once the muscle is activated on one side, it can only contract, and then the contraction forces will be transmitted to the epidermis on this side." I do not understand the sentence. Muscle activation leads to contraction, there is nothing to imagine here. Maybe you hypothesize that the muscles are attached to the epidermis such that muscle contraction leads to epidermis deformation?

Yes, four muscle bands are attached to the epidermis, as shown in Fig.1. The deformation does not concern only the epidermis but the whole embryo during the bending events. We have modified the sentence to avoid misunderstanding, the sentence change to “Once the muscle is activated on one side, it can only contract, and then the contraction forces will be transmitted to the epidermis on this side.” in line 71.

Ll 110: "However, it is less widely known that its internal striated muscles share similarities with skeletal muscles found in vertebrates in terms of both function and structure" Is it important for what you report, whether this fact is widely known?

Yes, it is our opinion.

Ll 112: "the role of the four axial muscles (...) is nearly contra-intuitive" Is it or is it not? If yes, why?

Yes it is. Muscles exert contractions, so compressive deformations. Their localization are along the axis of symmetry (up to a small deviation) so they cannot mechanically realize the expected elongation, contrary to the orthoradial actomyosin network.

However, elongation of the *C. elegans* is observed experimentally, so yes, we think the result contraintuitive.

L 116: "fully heterogeneous cylinder" What is this?

It means that the *C. elegans* embryo does not have the same elastic properties in different parts (or layers).

L 129: "will collaborate to facilitate further elongation" To facilitate or to drive? If the former, what drives elongation?

Contraction of muscles and actin bundles together drive elongation

Ll 141: "the deformation in each section can be quantified since the circular geometry is lost with the contractions" The deformation could also be quantified if the sections remained circular, right?

Yes. However, circularity is lost during each bending event.

Ll 151: "we need to evaluate the influence of the *C. elegans* actin network during the early elongation before studying the deformation at the late stage. So, the deformation gradient can be decomposed into: (...) where (...) is the muscle-actomyosin supplementary active strain in the late period" I thought you were now studying the early stage?

In this part, we are outlining how we can study the whole elongation (early and late), not just the early elongation stage. To evaluate the deformation induced by the first contraction of the muscles, we need to know the state of stress of the worm prior to this event, so we also need to recover the early period using the same formalism for the same structure.

L 160: "When considering a filamentary structure with different fiber directions" Which filamentary structure are you talking about?

Fig.3 B shows this model and the filamentary structure, which contains the actin and muscle fibers.

Ll 174: "When the cylinder involves several layers with different shear modulus 𝜇 and different active strains, the integral over 𝑆 covers each layer" I do not understand this sentence. Also, you should probably write 'moduli' instead of modulus.

This implies that when integrating over the whole cross-section S, we need to take into account each layer independently with its own shear modulus and sum the results.

L 176: "weakness of 𝜀" Do you mean \epsilon << 1?

Yes

Ll 178: "Given that the Euler-Lagrange equations and the boundary conditions are satisfied at each order, we can obtain solutions for the elastic strains at zero order 𝐚(𝟎) and at first order 𝐚(𝟏)." Are you thinking about different orders in an \epsilon expansion or the early and the late stages of elongation?

Answers: Different orders are considered only for the late elongation study, the early elongation is treated exactly so do not need a correction in \epsilon.

L 197: "fracture ablation" Please, define.

This is an experiment in which a laser is used to make a cut in a small-scale object of study and then the internal stresses are obtained based on the morphology of the cut, please see the Ref ‘Assessing the contribution of active and passive stresses in *C. elegans* elongation’. We have added this definition in line 200.

Ll 203: What motivated your choice of notations for the radii R_2'? The inner part of the cylinder is fluid? But above you wrote about a solid cylinder. Why should the inner part be compressible?

(1) We need to define the location of actin cables, which concentrate at the outer periphery.

(2) Our model is a hollow cylinder, and the inner part of the cylinder contains internal organs, tissues, fluids, and so on, so we consider it to be a compressible extremely soft material (Line 213).

Ll 212: "𝑟(𝑅) is the radius after early elongation." And during?

R is variable, r(R) depends on R but also on time t, it represents the radius of *C. elegans* embryos after the onset of elongation, i.e., after acto-myosin and muscle activities begin.

L 232: \tau_p is probably t_p?

Yes.

L 240: "quite simultaneously" Please, be precise.

In practice, it is difficult to define the concept of simultaneous occurrence unless there is rigorous experimental data to show it, but all we can get in the Ref ‘Remodelage des jonctions sous stress mécanique’, is that it occurs almost simultaneously, which we define as quite simultaneously.

Ll 246: "a short period" What does short mean? Why is it relevant?

From the experimental observations and data, we know that each contraction occurs very rapidly: a few seconds so we define a short period for one contraction.

L 263: "the bending of the model will be increased" Is it really the model that is bent?

Yes, the bending deformation predicted by the model, we have revised in line 266.

Ll 265: "we observed a consistent torsional deformation (Fig.6(E)) that agrees with the patterns seen in the video" In which sense do these configurations agree? I do not see any similarity between panels D and E.

Both show a torsion deformation.

L 267: "torsion as the default of symmetry of the muscle axis" I do not understand.

We discuss two cases in this research, one where the muscle follows the axis of the *C. elegans* in the initial configuration, and the other where the muscle has a slight angle of deflection, and we have added more information in the manuscript (line 270).

Ll 274: "Each contraction of a pair increases the energy of the system under investigation, which is then rapidly released to the body." Do you mean the elastic energy stored in the epidermis and central part of the embryo?

Yes, the whole body.

Ll 284: "The activation of actin fibers 𝑔𝑎1 after muscle relaxation can be calculated and determined by our model." Have you done it?

Yes, we can obtain the value of g_a1, and then calculate the elongation.

Ll 286 I do not understand, why you write about mutants at this place. Am I supposed to have already understood the basic mechanism of elongation? Why do you now write about the first stage?

I would like to show our formalism can model wild-type and mutant *C. elegans*, and the comparison results are good.

L 302: "The result is significantly higher than our actual size 210𝜇𝑚." How was significance assessed?Your actual size is probably more than 210µm.

Here, we have considered two situations, one is that the accumulated energy is totally applied to the elongation so that the length will be much larger than the experimental result of 210 µm, the length value that we have obtained by calculation. In the other case, we have considered the energy dissipation, which leads to 210 µm.

L 433: "where 𝜆 is the axial extension due to the pre-strained" Maybe ""where 𝜆 is the axial extension due to the pre-stress".

In our manuscript, we define the pre-strain, not the pre-stress.

L 438: "active filamentary tensor" Please, define.

Active filamentary tensor defines the tensor representing the activities of a cylindrical model composed of different orientations fibers.